# Modelling Consciousness within Mental Monism: An Automata-Theoretic Approach

**DOI:** 10.3390/e22060698

**Published:** 2020-06-22

**Authors:** Peter B. Lloyd

**Affiliations:** School of Computing, University of Kent, Giles Ln, Canterbury CT2 7NZ, UK; peter@peterblloyd.com; Tel.: +44-7856-675692

**Keywords:** idealism, consciousness, Hard Problem, automata theory, mental models

## Abstract

Models of consciousness are usually developed within physical monist or dualistic frameworks, in which the structure and dynamics of the mind are derived from the workings of the physical brain. Little attention has been given to modelling consciousness within a mental monist framework, deriving the structure and dynamics of the mental world from primitive mental constituents only—with no neural substrate. Mental monism is gaining attention as a candidate solution to Chalmers’ Hard Problem on philosophical grounds, and it is therefore timely to examine possible formal models of consciousness within it. Here, I argue that the austere ontology of mental monism places certain constraints on possible models of consciousness, and propose a minimal set of hypotheses that a model of consciousness (within mental monism) should respect. From those hypotheses, it would be possible to construct many formal models that permit universal computation in the mental world, through cellular automata. We need further hypotheses to define transition rules for particular models, and I propose a transition rule with the unusual property of deep copying in the time dimension.

## 1. Introduction

The modern mind-body problem, as formulated by Descartes [1], has expanded into the broad field of consciousness studies centred around what Chalmers [2] termed the ‘Hard Problem of consciousness’: even if we had a complete understanding of all physical processes in the brain, there would remain an explanatory gap between the physical workings of the brain and the operations of the conscious mind. Proposed solutions to this problem fall into three broad camps: Physical monism, which holds that only physical things are real; dualism, which holds that both the physical and mental worlds are real, and that minds cannot be reduced to purely physical systems; and mental monism, which holds that, ultimately, reality consists only of conscious minds. Mental monism is very much a minority position, but has been receiving growing attention owing to the failure of the more orthodox schools of thought. On the one hand, physical monism, in effect, denies the existence of the very thing we are endeavouring to explain, and whose actual existence is witnessed in every moment of our waking lives, namely consciousness. On the other hand, dualism has to suppose the presence of a physical substrate that is inherently incapable of direct observation and whose existence and non-existence are operationally indistinguishable. Mental monism is the only class of solution that does not suffer those metaphysical inconsistencies. One of the things that has discouraged interest in this theory is the so-called ‘bootstrap problem’. If there is nothing in reality except conscious minds, then the entire structure and dynamics of the mental world has to be explained by mental primitives, as there is no brain circuitry from which mental behaviour could be derived; and, as an extension of that derivation, all facts and laws of the physical construct must likewise be derived from mental primitives. This is, for example, one of the *desiderata* for idealism stated by Chalmers [3] (p. 356). I call this the ‘bootstrap problem’ by analogy with computer engineering: as a computer must ‘boot up’ or metaphorically pull itself up by its bootstraps, when it is switched on, so the entire observed universe must boot up from elementary mental entities. It is not immediately obvious how the bootstrap problem should be resolved, and the purpose of this paper is to argue for certain constraints on the family of formal models that could give an account of the bootstrap.

Lloyd [4,5] has given a detailed defence of mental monism and some corollaries of it, and those arguments will not be rehearsed here. The starting point in this paper is to suppose that mental monism is true (Premise 1 below), along with two corollaries—the non-spatiality of consciousness (Corollary 1) and the unity of subject (Corollary 2). I acknowledge these three propositions are contentious and require strong arguments in their defence. The author has offered such arguments in an earlier paper [5], and the reader is referred to that paper for details. In the present paper, the intention is to examine the implications of those three propositions for the modelling of consciousness.

Chalmers [3] has surveyed the species of mental monism or idealism that are now in circulation. They differ chiefly in scale, falling into three major categories: Micro-idealism, in which the universe comprises minds that are in some way closely related to microphysical entities; macro-idealism, which sees human minds as fundamental units; and cosmic idealism, which regards the universe as one big mind, with personal minds somehow partitioned off. The boundaries are by no means watertight, as Chalmers calls Berkeley a ‘macro/cosmic idealist’ since the Bishop divides reality into personal minds (human *et alia*) plus a big background mind (God) that takes care of everything outside the control of personal will. Idealism is closely related to panpsychism, which holds that every physical thing is conscious. (Confusingly, Chalmers uses “idealism” in a “broad” sense that includes panpsychism, which is normally regarded as a dualism rather than an idealism; and he uses “anti-realist idealism” for what is normally termed idealism *simpliciter*. I will stick to the more common usage here). What panpsychism and idealism do have in common is that they both assert that the basic units of concrete reality are conscious minds of some sort. The difference is that panpsychism also ascribes physical properties such as spatial location to those basic units, whereas in idealism the physical universe, including space, is a construct grounded in mental facts. According to Lloyd [5], if (and only if) we accept that physics is topic neutral, then the physical component of panpsychism drops away and the theory reduces to a variety of idealism. Hence, panpsychists who do not accept that physics is topic neutral, and who instead insist that consciousness is “all-pervading” (Shani [6]) or “ubiquitous” (Goff [7]) are not idealists because they ascribe spatial location to minds.

I will refer to the physical world as a ‘construct’, and a mind’s body in that construct as its ‘avatar’. The analogy with virtual-reality (VR) systems will be obvious.

Mental monism has had few modern supporters (e.g., Foster [8], Robinson [9], Hoffman [10], Kastrup [11]), while fewer have addressed the bootstrap problem (e.g., Hoffman, Kastrup), and I will consider the latter below, in the Discussion (Section 4).

In the field of formal modelling of consciousness, attention is normally given to models that are situated within physical monist or dualist (including panpsychist) philosophical frameworks. In such frameworks, the structure and dynamics of the conscious mind can be based entirely on the workings of the physical brain, either explicitly or implicitly. For example, Tononi’s integrated information theory [12], which is usually classed as a panpsychist dualism, places the main drivers for mental structure and dynamics in the information processing of the brain. Meanwhile, formal models within mental monism have been neglected. 

In fact, one frequently encounters ‘straw man’ objections to mental monism, which assert that it requires a *deus ex machina*, an intelligent and inscrutable deity that instigates and maintains the manifest world. For example, Bertini [13] (p. 123) states, “Scholars usually think that for immaterialism God is a typical *deus ex machina* of the Cartesian age”, and lists three examples of such thinking [14], (p. 297 et seq, [15]), [16]. Such deistic accounts are explanatorily delinquent as they presuppose an explanation more complex than the explanandum, and will not be considered here. Even when it is recognised as philosophically sound, mental monism is given little attention because of the difficulty of developing within it formal models of not just the mind but also of the physical construct.

The primary take-home message of this paper is that mental monism seems to entail certain constraints on possible formal models of consciousness, which are captured in Result 1 and Hypotheses 1 to 5 below. Additional Hypotheses 6 to 8 are not entailed by mental monism, but are speculative proposals for possible models. Although I do not present a full model here, it is suggested that the present analysis is a necessary preliminary to the development of such a model. 

### Structure of the Paper

This paper is structured as follows (see Figure 1). The fundamental Premise 1 (Section 2.1) is mental monism, with three corollaries: (1) that consciousness is non-spatial; (2) that there is at most one subject; (3) that all concrete entities in the model must be experientiable. (Premise 1, and corollaries 1 and 2, are taken from an earlier paper [5], Corollary 3 is new). A result drawn from this is that (Section 2.2) the mental world is discrete. The eight Hypotheses for formal modelling (Section 3) are as follows: Section 3.1 that the mental world follows natural laws; Section 3.2 that it includes free will; Section 3.3 that it is built up from mental primitives (‘experientiae’) each of which is both experiential and volitional; Section 3.4 that mental space is built up from those primitives; Section 3.5 that mental individuation and intercommunication can be understood in terms of those primitives; Section 3.6 that the mental world is built up from a small palette of experiences; Section 3.7 that a minimal set of operators apply to the experientiae; and Section 3.8 that the experientiae transition stochastically. The first five Hypotheses form a core that I propose should apply to any formal model of consciousness within mental monism, while the last three are more speculative.

These Hypotheses under-determine the basic rules of the mental world. In particular, a key question concerns the transition rule obeyed by mental primitives when regarded as units in a cellular automaton. Traditionally, such rules are set by the investigator, on the premise that there is a substrate in which the cellular automata are implemented. I argue that that would be an inelegant solution in the present case and unlikely to be fruitful. Instead, an unconventional kind of transition structure is considered to avoid arbitrary transition rules, namely deep temporal copying (see below, Section 3.7); and, for further simplicity, the transitions are hypothesised to be stochastic (Section 3.8).

Finally, I discuss the relationship between this model and those proposed by Hoffman (Section 4.1) and Kastrup (Section 4.2), and discuss the possible connection between this model and cytoskeletal cellular automata, which Penrose and Hameroff [17] have suggested as physical correlates of consciousness (Section 4.3). The relationship between the present approach and ‘digital physics’ is also discussed (Section 4.4).

## 2. Fundamentals

### 2.1. Premise 1: Mental Monism

The fundamental premise of this study is the philosophical theory of mental monism (also known as ‘subjective idealism’ or ‘mentalism’). This is the doctrine that reality is wholly mental, and that what we take to be the physical world is a derived construct. This doctrine was proposed by George Berkeley in the Eighteenth Century [18], and articulated in modern terms by Foster [8], Robinson [9], Lloyd [19,20], and Pearce [21]. Lloyd [5] provides a proof of this doctrine, roughly equivalent to those of Foster and Pearce, and following the original reasoning of Berkeley. I acknowledge that mental monism is highly contentious, but I will not rehearse the argument for it here as I have already covered it in the foregoing reference. The task addressed in this paper is somewhat narrower: taking mental monism as a premise, how might we formally model the conscious mind? 

**Premise** **1.**
*(Mental Monism): Reality consists just of a set of minds U = {M*
_0_
*, M*
_1_
*, M*
_2_
*, …}.*


Here, a mind is represented as a time-varying structure M(t) = <S, C(t), R(t)> where S is the fixed subject, C(t) is a set of conscious experiences, R(t) is a set of binary relations between experiences; and t is time.

I will now look at three corollaries of this premise. These are not counted among the Hypotheses of this paper, as they are directly entailed by Premise 1, which I am regarding as a given. Premise 1 is defended in [5] (Section 3) and Corollaries 1 and 2 in [5] (Section 2.7).

It immediately follows from mental monism that physical spacetime has no mind-independent reality and cannot serve as a ground within which minds exist. So, not only do we have to model the conscious mind without the substrate of the brain, we have to do so without a spatiotemporal medium in which to hold the conscious experiences.

An objection to this is that, under the special theory of relativity, time and space cannot be separated. If an event is in physical time then it must also be in physical space. Lloyd [5] (Sections 4.1 and 4.2) gives a detailed rebuttal of this objection. The conclusion from that rebuttal is that mental events are in neither physical space nor physical time, and that physical spacetime must somehow be constructed from consciousness. Likewise mental space (‘mindspace’) must be constructed (see below, Section 3.6). Mental time is taken as a primitive insofar as it describes the succession of moments of experience, but it has no direct connection to physical time. The non-spatiality of consciousness plays a central role in the subsequent Hypotheses.

**Corollary** **1.**
*(Non-Spatiality): Minds do not have physico-spatial relationships.*


Furthermore, the conventional individuation of minds and subjects by means of their spatial separation is no longer feasible in mental monism as minds are not embedded in space. Let me pause to clarify the terms ‘subject’ and ‘mind’. The former is the agent of acts of experiencing and volition; the latter is the combination of subject and contents of consciousness (experiences and their relations). The two terms are sometimes conflated, but the conceptual distinction is well established. Hume [22] wrote, “For my part, when I enter most intimately into what I call myself, I always stumble on some particular perception or other… I never can catch myself without a perception, and never can observe anything but the perception”. He concluded that the subject as agent did not exist, and proposed instead the ‘bundle’ theory of selfhood. James [23] (p. 304) agreed with Hume’s observation but not his conclusion: “the existence of this Thinker would be given to us rather as a logical postulate than as that direct inner perception”. Wittgenstein [24] (pp. 66–67) understood the subject as a peculiar usage of language. In the present paper, nothing hangs on whether the subject is an ontologically real thing, or only notional: it plays the same role in the formalism. 

Conventionally, it is held that two minds are distinct by virtue of their sitting inside distinct brains, which are in different places. Even if two minds had absolutely identical content, personalities, behavioural traits, abilities, memories, thoughts, emotions, and so on… if they were sitting in different brains, then the conventional view is that they would be regarded as different minds. If the two brains were molecule-for-molecule identical, and given identical inputs, the contents of the minds would remain aligned, but we would still say they are two distinct minds because they were in two places. Furthermore, the conventional view is that the subjects of those two minds inherit their individuation from the individuation of their respective minds. This criterion of individuation has to be re-thought radically in mental monism, because neither minds nor subjects are in space.

Thus two minds (as distinct from subjects) can be individuated by their content, but not by their position in space (which they do not have); whereas the subject has neither content nor position. Therefore, without containing content, and without being anchored in space, subjects simply have no individuation. There is therefore only one subject. (This conclusion might seem a large and surprising leap, but a longer defence of it is given by Lloyd [5] (Section 2.7). Although Western thinking may view this conclusion as odd, the Hindu school of Advaita Vedanta has held it as a central tenet since the Eighth Century CE: “Thou art Brahman”). Hence:

**Corollary** **2.**
*(Unified subject): There is at most one subject.*


Finally, we have to address the modality of modelling. When dealing with an abstract system, we are at liberty to posit the existence of any entities that make the model workable and tractable. For example, in particle physics, Feynman posited ‘virtual particles’, which are needed for his diagrams to work as part of the theoretical apparatus, but they can never be observed. This is a legitimate move, as particle physics is abstract. There is no independent criterion of concrete existence that the virtual particles must meet. According to mental monism, however, the only concrete things that exist are conscious minds. This furnishes a stringent extra criterion that any model of consciousness within mental monism must meet. Such a model can refer only to concrete entities that are experiential. The modeler is not at liberty to posit novel entities that cannot be experienced, even though they might make for a formally more elegant model.

We will see an example of this below when we look at discrete versus continuous experiences. Since continuous experiences cannot be perceived, mental monism disallows them as concrete existents, and therefore they cannot be posited as concrete entities in the model.

Abstract, as opposed to concrete, entities are not problematic. Thus the structure of the 3-tuple <S,C,R> and the relations R are abstract artefacts of the model: they cannot be experienced and are not concrete existents. Whether the subject S can, in some sense, be experienced is a moot point but, as noted above, the model does not hang on this.

**Corollary** **3.**
*(Concrete modelling): All concrete entities that form part of the model of consciousness (within mental monism) must be capable of being experienced.*


### 2.2. Result 1: Mental Discreteness

A fundamental requirement in this approach is that the conscious mind is a discrete system, and therefore lends itself to the methods of automata theory. 

As the conscious mind is not amenable to third-person observation, we must rely to a significant degree on introspection and even intuition. To some people, including the present author, it seems self-evident that the conscious mind is a discrete system, and I am grateful to an anonymous referee for emphasising that others have an intuition diametrically opposite to this. Given that the rest of the paper hangs on this result, its defence is expanded as follows: motivation; literature overview; the illusion of continuity; counter-arguments; inobservability of an experiential continuum; formal argument. If the conscious mind were a continuous system, rather than a discrete one, then it would require a different class of model from the one proposed in this paper. 

#### 2.2.1. Motivation: The Finitude of Cognition

The phenomenal field comprises discernible parts, which have spatial, temporal, structural, and qualitative relations, and are subject to limitations of acuity in space, time, logical composition, and quality. For example, I can perceive the contents of my visual field as a two-dimensional assemblage of discernible patches of different colours and brightness, and other features, with limits of resolution in mental time and space, limits of association with other mental content, and limits of discernible gradations of quality. In operational terms, the evidence for this can be outlined as follows. First, between any two points in any sensory field (visual, tactile, auditory, or proprioceptive), there is a finite number of discernible positions. Second, between the start and end of any interval of time, a finite number of moments can be discerned. Third, from any given experientia, there is a finite number of associated memories. Fourth, along any gradient of quality (brightness, redness, sweetness), there is a finite number of discernible levels. We are concerned here, not with what the specific numbers are, but only with the fact that these intervals (spatial, temporal, associative, and qualitative) are not infinitely divisible, and that therefore a discrete system is a legitimate model to try out.

For the removal of doubt, this argument refers to the resolution of the mental sensorium, not to the acuity of perception of physical stimuli by bodily organs. An analogy might help. Suppose I have a digital camera with 7000 × 7000 pixels. With one lens, I might have a resolving power of a quarter of an arcminute, with another lens I might be able to resolve only half an arcminute. The digital resolution remains the same forty-nine megapixels, but the resulting picture is quite different. A human eye has a resolving power of about 1 arcminute, and the retina about 126 million light-sensitive cells, concentrated in the fovea. Roughly speaking those parameters define the acuity of the eye. The composition of the mental visual field is conceptually fundamentally different from either the optical image falling on the retina, or the pattern of electrical signals sent down the optic nerve, but is easily conflated because evolution has matched them. If the structure of mental visual field were much finer, then there would be a redundancy, and the mental field might seem blurred; if the eye had significantly more resolving power then it would be wasted as the conscious mind would not have the structure to contain the extra information. The fact that the neural representation of the optical image is discretised in the axons in the optic nerve is not an argument for the discreteness of the mental visual field. Rather, the argument for the mental discreteness stems from an introspective consideration of conscious experience itself.

#### 2.2.2. Literature Overview

Hameroff and Penrose [17] have proposed discrete mental time as a concomitant of their model of the physical correlates of consciousness, and have surveyed the literature on this question. They trace the notion of ‘moments of experience’ to early theoreticians such as James [23] (p. 631) and Stroud [25,26]. In Buddhist thinking, discretisation of mental time has a much longer history. Von Rospatt [27] gives an extended account of the historical development of the doctrine of momentariness (*kṣaṇikavāda*) in Abhidharma and early Yogåcåra in the Fourth Century CE. VanRullen and Koch [28] surveyed neuropsychological evidence for discrete or continuous perception but concluded only, “It seems surprising that such a fundamental question as whether conscious perception occurs in discrete batches or continuously has not been definitely answered one way or another”. More recently, Herzog, Kammer, and Scharnowsk [29] have argued strongly for a discrete model of perception. This, however, remains a contended claim in neuropsychology, and White [30] has commented, “it is not even clear what sort of evidence would demonstrate the occurrence of discrete frames”. Tee [31] has analysed the question from Shannon’s classical information theory and concluded, “Going forward, we believe that the correct research question is no longer that of continuous-versus-discrete, but rather, how fine grained the discreteness is”.

A severe methodological difficulty in these neuropsychological investigations is that consciousness is not open to direct, third-person observation, and there is an inferential gap from any conclusions about neural processes’ being discrete or continuous and mental processes’ being discrete or continuous. Although the Buddhist analyses lie outside third-person science, they consistently lie within the domain of consciousness, and might be considered more pertinent. 

#### 2.2.3. The Illusion of Continuity

The belief that mental experience is continuous is often said to be more intuitive than its opposite, the belief that it is discrete. For example, vanRullen and Koch [28] wrote, “a continuous translation of the external world into explicit perception [is] more intuitive and subjectively appealing”; Herzog, Kammer, and Scharnowsk [29] wrote, “We experience the world as a seamless stream of percepts”; White [30] wrote, “subjectively, conscious perception is smooth and continuous”; and Tee [31] wrote “Historically, most analyses assume a continuous representation without considering the discrete alternative”; and one of the anonymous referees for this paper wrote, “our experience of time is most definitely that of continuous flow”.

Whence this intuition? It may stem from the mind’s notional gap-filling. Suppose that conscious experience really were discrete in time. Since, *ipso facto*, there is no gap between the successive moments, there is no sensation of a gap to be experienced. (If there were an experience of gaphood between two successive moments of experience, A and B, then that would simply be another, intermediate moment of experience, C, contradicting the premise that A and B were successive). On the other hand, if the two successive moments of experience have a barely perceptible difference, then there is no awareness of a jarring change. We are already very familiar with this phenomenon when watching a film at 24 frames per second. If the moments of experience were to succeed each other at, say, 75 times a second, as Buddhist doctrine suggests, then there seems to be no reason to think we would ever notice the discreteness of experience, unless we undergo extensive training in Vipassanā meditation [32]. 

It is therefore suggested that the intuition of temporally continuous experience may be an illusion due to the gapless succession of barely distinguishable experiences, and that the intuition of continuous experience in other dimensions may likewise be illusory. 

#### 2.2.4. Counter-Arguments

There are some counter-arguments to discrete mental experience. It could be suggested that the conscious mind actually inhabits a continuous mental space, but that the contents happen to be discrete because they are produced in lock-step with discrete neural systems. A related argument is that the sensorium is actually continuous but we lack the skill to discern this continuity as we become vague and confused when mental content is too tightly packed. There is inherently no evidence for this, as it posits experiential content that we are not aware of. Another counter-argument is that the apparent decomposition of the sensorium into finite numbers of discrete elements is not real but a methodological artefact, it is produced by the very method of trying to see what reality is made up of. According to this postmodern perspective, experienced reality has no genuine constituents, but we can create the impression of composition by performing certain operations on our mental content, so as to yield a decomposition. For example, on this view, the experience of a sticky toffee pudding is an unanalysable sense datum, and the process of decomposing that experience into specific flavours, smells, textures, heat feelings, sensation of solidity, softness, and stickiness, and moments of change thereof, does not reveal the pre-existing constituents of the sense datum but only generates new sensations in isolation. Moreover the postmodernist would maintain that there are multiple, equally legitimate modes of decomposition, hence the decomposition cannot reveal a prior, objective composition, but merely produces a subjective derivative. This speaks to a deep methodological divide between the rational-scientific enterprise and the anti-scientific programme of postmodernism. The main effective argument against this line of thinking is utility: if we find that independent decompositions converge on a common model, and this model yields testable hypotheses that are found to be confirmed, then we can at least conclude that the model, *qua* model, is valid. The only way to make such an assessment is to attempt to model consciousness in this analytic manner. 

Approaching the matter from another angle, some have argued that, since physics has to be grounded in consciousness, and physics has a lot of continua, it might be more natural to have a continuous rather than discrete model of consciousness. In virtual reality systems, however, digital computers can simulate the continuous world of physics with as much fidelity as the computing speed and sophistication of the software permits. The Cook-Karp Thesis shows that anything computable by an analogue computer is also computable by a discrete automaton, provided that it has the power of a Universal Turing Machine. Therefore, continuousness in the physical construct does not require continuousness in the mental systems that support it, and therefore it is not an argument against a discrete model of consciousness. 

#### 2.2.5. Inobservability of Continuous Perception

What could possibly count as evidence for continuous experiences? No direct observations could ever reveal them. The set of all conscious experiences that have ever occurred is finite. (This makes the reasonable assumption that humanity has existed for a finite time, as per standard accounts of human history). Even if humanity carries on forever, there would be only a countable infinity of observational data. There is no way for us to perceive a continuum. 

A counter-argument is that mathematicians have, since the invention of calculus in the Seventeenth Century, had no difficulty in handling uncountable infinities of points, lines, and planar surfaces, and therefore similar techniques would equip us to deal with continuity in the mental world. Following up that line of thinking, we may anticipate a *reductio ad absurdum* counter-argument by analogy between mental and physical distances, as follows. Each occasion of measuring physical distance is a discrete event, therefore we can carry out only a countable number of measurements of distance, therefore distance cannot be continuous. That *reductio* argument, however, is derailed by the fundamental difference that physical space is an abstraction that is wholly and exclusively defined by its axioms, whereas mental space is a concrete phenomenon. In mathematics we are at liberty to define ℝ as the real line, and lo! it has an uncountable infinity of points. We do not have to observe them, as Dedekind’s axioms guarantee that they are there. In consciousness, what the mental field contains is a brute fact and we do not axiomatically construct it. If it is proposed that the mental field contains an uncountable infinity of experiences, then there had better be empirical evidence for their existence, since we cannot conjure them axiomatically. Empirical evidence for an infinite sea of experiences is not possible: we can take cognisance of only a finite set of discrete experiences, which would appear the same if they were all there is or if they were sampled from an infinite set of experiences. Hence, the claim that the mental field is continuous, not discrete, is not falsifiable, and therefore not a Popperian scientific hypothesis

#### 2.2.6. Formal Argument

The basic tenet of mental monism denies concrete reality to anything that is not a conscious experience (see Corollary 3 in Section 2.1 above). Since a subject has only a limited ability to differentiate mental experiences, two experiences that are not differentiable are not experienced as two distinct experientiae and so, by mental monism and Corollary 3, do not *exist* as distinct experentiae. For example, suppose two specks A and B in your visual field have a barely distinguishable separation. According to the continuum notion, there should be a third experience C between them (and not just C but an uncountable infinity of intermediate experiences). Yet, *ex hypothesi*, there is no discernible distance between them, so either C is just A, or C is just B; in either case C is a not a distinct experience—which contradicts the premise that C is another experience. Precisely the same reasoning applies to all other dimensions of mental experience. For example, suppose A and B have a barely distinguishable distance in time. According to the continuum notion, there should be infinitely many experiences between A and B, such as C. Yet if A and B are barely distinguishable then C is either indistinguishable from A or indistinguishable from B: either way, clearly there cannot be any distinct experiences between A and B. Hence we are led to:

**Result** **1.**
*(Discreteness): The phenomenal content of a conscious mind is a discrete system.*


Therefore, in the representation M(t) = <S,C(t),R(t)>, where S is the fixed subject, C(t) is a *countable* set of conscious experiences, R(t) is a *countable* set of binary relations between experiences; and t is *discrete* time. By introspection, I would expect C and hence R to be finite, but the premise of mental monism implies only that they are countable, not that they are necessarily finite.

Understanding the mind as a discrete system opens up the possibility of treating it formally as a cellular automaton. In a classical cellular automaton, we have a set of unit automata, each of which has fixed spatial relations with its neighbours, and each possesses a number of states. In that conceptual framework, the unit automaton and its states are two ontologically distinct things; but, as we will see below, mental monism will require a more parsimonious model than that. 

## 3. Hypotheses for Possible Models in Mental Monism

My aim is to examine the constraints that mental monism imposes on formal models of consciousness by means of eight minimal Hypotheses (Section 3.1, Section 3.2, Section 3.3, Section 3.4, Section 3.5, Section 3.4, Section 3.7 and Section 3.8) about the behaviour of the mental primitives. I will argue that the first five hypotheses form a core that are necessary, as they are either pre-requirements of a workable model (Hypotheses 1, 2) or entailed by mental monism (Hypotheses 3, 4, 5), while the remaining three are speculative (Hypotheses 6, 7, 8). This set of Hypotheses points toward mental systems that can embody cellular automata, which are known to be capable of implementing Universal Turing Machines, which can compute any computable function. The motivation for doing so is that once we have established that a mental system is capable of universal computation, then we have a ‘proof of concept’ that it is possible, in principle, to construct *purely mental* mechanisms for (a) psychological functions such as memory, body image, cognition—without deriving these mental functions from the workings of the brain; as well as (b) the whole manifest world as observed through the senses—without deriving it from a mind-independent environment.

The motivation for the whole exercise is, of course, that the philosophical theory of mental monism denies the existence of the brain and the rest of the physical world, and can be viable only if there is a constructive route from mental primitives to the mind and everything observed with it.

### 3.1. Hypothesis 1: Naturalism

We should seek to account for the structure and dynamics of the mental world with a model that is nomologically constrained. For otherwise, without laws that govern the mental world, we have either chaos or magic, and not a usable model at all. As the mind is a dynamic system, the most natural form of nomological constraint is a causal relation, in which the successive states of a mind are largely driven by earlier states. This need not be a total determinism: a causal connection in which some state changes are nondeterministic could still be nomological enough to yield the stable behaviour that we actually observe in the mind. 

By the way, teleological models are not considered here: I hypothesise that reality is not, at its fundamental level, teleological.

Furthermore, we should seek a model in which cause-and-effect is mediated through direct connection and not a spooky disengaged action. The first thing we notice about the mental realm is that it is compartmentalised into minds, each of which is private and interacts with other minds through specific input/output ports. I am going to refer to this as ‘naturalistic’ because any alternative—that is, behaviour that is predominantly uncaused, or caused by events in other minds—would undermine our everyday observations of how our minds work. It would also face the so-called navigation problem [20] (Section 4.3.1): if a mental event E_1_ is to cause another such event E_2_, and if E_1_ and E_2_ have no direct or indirect contact, then how does the causal influence navigate from E_1_ to E_2_ and not go by mistake to, say, E_3_? Unless the causal influence is directed in a way that is random, or magical, we would have to posit a complex broadcast-and-recognition mechanism behind the scenes that ensures the causal influence is received in E_2_, not E_3_. We cannot rule out such a thing, but having Ockham’s razor in hand, the proper course is first to study—and, if necessary, eliminate—the simpler models rather than jumping straight to a more complex model.

One of the anonymous referees argued that it is incongruous to impose a principle of mediated causation on the model of consciousness, when that model must eventually ground physics, which is nonlocal.

In physics, however, we do find the universal operation of local causation, which is the physical counterpart of mediated causation—namely that cause-and-effect works between events that are in contact, or have a chain of contacts, either through material objects or through immaterial fields. Even quantum physics has no nonlocal causation, nor the nonlocal transfer of energy or information. In Aspect’s famous experiment [33], the particles remain entangled no matter how many miles separate them, yet nothing is transmitted between them. The No-Communication Theorem [34] shows the impossibility of superluminal communication or energy transmission, which entails that—between events that are situated within spacetime—causation must be local, even in the face of quantum entanglement.

Contrary to a common misunderstanding, orthodox quantum mechanics allows only nonlocal correlation, not nonlocal causation. For example, in Aspect’s experiment, when measurements are made on two particles that are entangled, but spacelike separated, then their states are found to be correlated. In the popular imagination, it is supposed that the first particle to be measured will nonlocally determine the state of the other particle. Not so. In relativity, the sequence in which spacelike separated events occur depends on the inertial reference frame of the observer, and there is therefore no absolute fact of the matter which event occurs first. As time and space are inextricably linked in spacetime, if something is nonlocal in space, then it must be nonlocal in time, so the ordinary concept of causation, which goes forward in time, is inapplicable. How can we say that a nonlocal event E_0_ caused some event E_1_ if it is indeterminate whether E_0_ preceded E_1_ or vice versa? (There are unorthodox interpretations of quantum mechanics that posit global hidden variables. Whether some novel concept of nonlocal causality might be devised for them is a moot point, but anyway this would not threaten the No-Communication Theorem, which is all that I am referring to here).

When it comes to mental events, although the conscious mind is not situated in physical spacetime (Corollary 1 above), the model proposed here has a notion of mental time (albeit unaccompanied by any prior notion of mental space), and it is therefore meaningful to speak of one mental event causing another. Hence it is meaningful to state the first Hypothesis:

**Hypothesis** **1.**
*(Naturalism): The time evolution of a mind is internal, M(t + 1) ∈ Ψ(M(t)), where the right-hand side is a set of possible mental states (possibly a singleton).*


That is to say, given a state of a mind M at time t, a function Ψ applied to M determines the set of possible states at the next moment. In the real world, of course, a mind is affected by sensory inputs, but those inputs can affect the operation of the mind only after they have arrived ‘inside’ the mind. I will discuss below (Section 3.5) the problem of how inputs get into the mind, but for now let us just note that the standard physical-realist route is not available in mental monism. Specifically, the physical-realist route is as follows. In any model where the mind supervenes on the brain, a physical stimulus arrives on the doorstep of the brain (for example, light waves arrive at the retina, and the normal local physical processes—that is, signal propagation along nerve fibres—carry that stimulus into the brain, where the supervenience is supposed to happen). In such a framework, mental input piggybacks on the physical proximity of sensory stimuli to the neural correlates of consciousness. In mental monism, there is no space, hence no proximity, hence input into this substrate-free mind must be modelled by a route quite different from the physical-realist route. This is examined below in Section 3.5.

### 3.2. Hypothesis 2: Volition

For some people, an hypothesis of volition is somewhat extravagant and should not be assigned to a minimal set of hypotheses. I suggest, however, that volition is, in fact, a feature of the conscious mind that is universally apprehended, but a genuine nondeterminism cannot be modelled by a deterministic system, therefore it must be brought in as an hypothesis *ab initio*.

The notions of determinism and nondeterminism are always relative to a particular transition function in a particular system. To be pedantic, we could say that the volition ‘determines’ the outcome but that is not the normal usage. Rather, the fixed transition function Ψ ‘determines’ either a singleton outcome or a set of possible outcomes, and volition selects one of those outcomes.

The notion of volition is distinct from that of random chance, which differs from bare nondeterminism by requiring a particular probability distribution. We know from personal introspection that we have volition; but whether anything happens by random chance is a moot point, and outside the scope of this paper. 

**Hypothesis** **2.**
*(Volition): The time evolution of the mind involves a nondeterministic component. That is, the time evolution, Ψ, may yield a non-singleton set of outcomes, |Ψ(M(t))| ≥ 1. Free will determines which outcome is selected: M(t + 1) = ω(Ψ(M(t))).*


### 3.3. Hypothesis 3: Unitary Experientiae

We now turn to the transmission of causal effect, and hence of information, between minds, and between parts of a mind. In the physical construct, we are accustomed to the central role that proximity plays in the transmission of causation. The classic example is a billiard ball striking its neighbour and imparting kinetic energy; a photon striking the retina and imparting its electromagnetic energy; a gravitational field gripping an object and pulling it in. Even nonlocal quantum-mechanical phenomena do not involve the observable transmission of energy or information from A to B instantly without passing through the intervening space. In mental monism, this role of proximity cannot be utilised, as there is no concept of proximity, because (by Corollary 1) there is no space. So, we have to rethink the mode of causal transmission in mental monism.

At a first inspection, it seems that the mind comprises both experiences (‘ideas’ in the Seventeenth Century term used by Locke and Berkeley) and acts of volition—the latter manifested both as changes in internal thoughts and imagery, and as motor actions. That apparently basic conceptual separation between experiences and volitions is, I shall now argue, not possible in mental monism. Instead, I shall argue that the mind comprises ‘unitary experientiae’ that are both experiential and volitional in nature. This counter-intuitive conclusion is an unexpected consequence of the non-spatial mode of causal transmission in the realm of consciousness.

Let us consider the following two suppositions, and then show that they lead to absurdity, and must therefore be rejected. (This argument is situated within the context of mental monism, of course).

**Supposition** **1.**
*A conscious mind comprises elements of two, mutually exclusive classes: experiences and volitions.*


**Supposition** **2.**
*A volition in one mind produces experiences in another mind through an intermediate mechanism.*


We shall consider in turn the two classes of experience, ‘outward perceptions’ and ‘inward imaginations’. As discussed by Lloyd [5] and Pearce [21], ‘outward’ is used in its everyday sense and not in a metaphysical one that would imply a mind-independent object of outward perception.

**Outward perceptions.** Consider, first, perceptions that occur in your mind due to the manifest world. For example, you look up at the sky and have an experience of phenomenal blue. By Hypothesis 1 (Naturalism), this is not random noise but the product of some process. By Premise 1 (Mental Monism), that process must be the activity of a conscious mind, since there are no other entities that could be producing the phenomenal blue. Following Lloyd [19], I will use the general label of ‘metamind’ for the mind(s) responsible for natural phenomena, that is, all actions that are not acts of personal volition. This term is preferred over Berkeley’s “God” or Śaṃkara’s “Brahman” as it avoids religious connotations. (The metamind might correspond to part of Hoffman’s [10] community of ‘conscious agents’). Therefore, we may say that your sensation of phenomenal blue in the sky is produced by a volition of the metamind. At first, it may seem that the metamental volition and your phenomenal experience are two distinct things, and we must therefore suppose there is some intermediate mechanism that transmits the causal influence from the volition to the perception, for Hypothesis 1 (Naturalism) prohibits any spooky unmediated influence. What could be the nature of that mechanism? By Premise 1 (Mental Monism), reality comprises only conscious minds. So, the putative mechanism that conveys causation from mind A to mind B, must be a third mind C. But then we have to ask how the causation gets from A to C. Repeating the foregoing argument, we must infer another mind, D, that conveys causation from the volition of A. So we have an infinite regress, which is absurd.

We might be tempted to think there could be an articulation between the agent and percipient (in our example, the agent is the metamind and the percipient is your personal mind). For example, could it be that the agent is ‘adjacent’ to the percipient in some sense, and just rubbed up against it in some causally efficacious way? No, because, by Corollary 1, we have no concept of spatial distance between minds, hence no concept of adjacency of minds. So the agent cannot affect the percipient through ‘contact’.

By this process of elimination, the only concept left on the table is that the agent overlaps with the percipient. So, the metamental volition occurs inside the percipient mind. Thus Supposition 2 must be dismissed, and we can now focus on Supposition 1. 

**Inward imagination.** This also brings us to the second form of action, that of an intra-mental act of volition, for example, where you imagine something. We are familiar with our own imaginings, but by the foregoing argument, even outward perception must be understood as an intra-mental act of volition (as the metamind’s volition to produce the blue patch must occur inside your mind). The following remarks therefore apply to both cases, outward perception and inward imagining.

Now we have an analogous logical conundrum to the one that was addressed above. If we suppose that the causal influence between the volition and the experience (within a mind) is conveyed by an intermediate entity then the intermediary can be only another volition or an experience, which just leads us to the absurdity of an infinite regress. If we were to suppose that the proximity of the volition to the experience would enable a causal power to be transmitted, then we are stuck again as there is no primitive intra-mental space, any more than there is any inter-mental space. Now this process of elimination leaves us only one option on the table, that the volition is inside the experience. But these are elementary things, so we are forced to conclude that the volition and the experience are the same thing. I shall use the term ‘experientia’ for this unitary entity that is both volition and experience.

Furthermore we must say that every experience is also a volition (for, otherwise, what is there to cause it?) and every volition is an experience (for, otherwise it would not produce any result). 

This brings us to a simple picture in which reality comprises a countable set of minds, and each mind comprises a countable set of experientiae, each of which is both volition and experience, and one mind communicates with another by executing a volition within the recipient mind. The agent mind and recipient mind each may be personal minds or the metamind. 

**Hypothesis** **3.**
*(Unitary Experientiae): The contents of a conscious mind is a set of unitary experientiae. Each experientia is both volition and experience and, conversely, every volition and every experience is an experientia.*


**Result** **2.**
*Communication between two minds is achieved by executing a volition inside the recipient mind.*


I admit that these two propositions seem odd, but I submit that this is just because we are so accustomed the spatially grounded causality of physics, and we have to shift to another way of thinking, one that is closer to that used in computer science. For example, if program A is to communicate data directly with program B, then they can read and write in a shared area of memory, which is inside both programs. (In practice, programs can also communicate by reading and writing in a file that they both open, but then the analogy breaks down as there are two kinds of thing, programs and files, whereas mental monism has only one kind of thing, namely minds).

### 3.4. Hypothesis 4: The Construction of Mental Space

Our experientiae present themselves as in a space-like configuration, the ‘mindspace’. Mental space cannot, however, be a primitive in mental monism because, by Corollary 3, we can admit only experientiables and mental space is not itself experienced. Therefore, those spatial relations must be constructed somehow, rather than being primitive. This section therefore examines what kind of models could account for the construction of mental space.

Within a physicalist or dualist models, mindspace could readily be explained by referring to the underlying brain structure, which is spatially isomorphic to afferent nerve endings. Within the framework of mental monism, however, we do not have that neural substrate, and mental space must be constructed out of mental activity only.

On a naïve view, we may be tempted to conceive of the sensorium as being like an inner cinema screen. A moment’s reflection shows that this is untenable as it implies an homuncula watching the inner cinema screen, who in turn has her own inner cinema screen, *ad infinitum*.

On a first inspection, it would appear that there is a persistent body-image, consisting of a finite number of ‘places’, each of which can contain sensory experiences of a type specific to that section of the sensorium. For example, my skin appears to form a two-dimensional space, topologically equivalent to a sphere, divided up into places each of which can harbour sensations of heat, texture, pressure, and so on.

It is therefore tempting to envisage the manifold of mental places as being like an array of pixels, but that does not sit well with experimental data. Since the work of Hubel and Wiesel [35], it is apparent that the ingredients of the visual sensorium include lines and movements as well as points. Nevertheless, we can use the term ‘places’ for these spatially differentiated components, as long as we bear in mind that they are not just point-like ‘mental pixels’.

With regard to outward-facing senses, this architecture of mental space is self-evident: your visual experiences occur in a two-dimensional field of vision. You automatically project what you see into the constructed three-space surrounding your body, but your visual field is not itself three-dimensional, it only has associations of three-dimensional structure attached to it. Extending out from the visual field is the tactual field. Although this is a different sensory modality from sight, it is situated in the same mental space, a fact that is trivially established by touching your eye: you see the finger approach the pupil, and then you feel the eyeball embedded within the tactual field. Likewise hearing, although the source of a sound is more diffusely spread around the circumpersonal space, the auditory experientiae are situated in the same mental space as vision and touch. This is trivially established when you put your fingers in your ears to block out a sound. We can go through all the senses in like manner: taste and smell are situated in the body-image at the buccal and nasal passages. Proprioception locates qualia of limb position in intermediate positions within the topological boundary of the body image. Pain sensations such as nausea, toothache, headache are likewise situated inside the body image. Thoughts are experienced as if contained within the body-image of the head and, if articulated will be ‘heard’ as vocal sounds in the ‘inner ear’ (that is, projected to the positions of the ears) or ‘seen’ as words in the ‘inner eye’ (that is, projected to the visual field). Braille users project words to tactile sensations in the fingers. Emotions are felt in the head or in organs of the body (for example, fear in the intestines). Although thoughts and emotions carry a propositional freight, our immediate awareness of them is constituted by qualia. You know that you have a certain thought only by virtue of the auditory, visual, or other sensory markers that indicate the thought. (Maybe, when reading this paragraph, your heard the words “That’s rubbish!” in your inner ear? Or saw yourself writing “Thoughts have no qualia”?) That is not to say that hearing a certain sequence of words in the inner ear is, on its own, constitutive of the thought. (E.g., in the first example, what does “That” refer to?) Rather, the words form part of a semiotic constellation of experiences, perhaps involving images from other modalities. Whether that constellation of sense-images is itself constitutive of the thought is a question I will address in the next section. Here, however, my conclusion is that no sensation is wholly without a spatial relation to the rest of the mental contents. That is, all sensory mental content is situated in the mental space of the body image.

What, precisely, are those constructed spatial relations in mindspace? Recall that, in mental monism, there is no given physical space within which experientiae could relate spatially. Mental spatial relations must therefore be wholly constituted by some pre-spatial aspect of the mental structure and dynamics. So, we must inquire what precisely it means to say, for example, that a particular green patch in the visual field is ‘below’ a particular blue patch. The naïve answer would be that the two patches are situated at two positions in a spatial medium, and the vertical coordinate of one position is greater than that of the other. That naïve account, however, will not work in mental monism because there is no prior space in which experientiae are sitting. Suppose there were such a space. Then it would have to be an experientia. And then we would have to explain what it means for the blue-patch experientia to be in a certain position in relation to the space experientia. Which would require a further space in which the space experientia sits, which begins an infinite regress. For sure, whatever it is that constitutes spatial relations must comprise experientiae, as those are the only building blocks of the mind allowed by mental monism. But they could not be static, for example, an ‘aboveness’ quale and a ‘belowness’ quale, as that would underdetermine the direction of the relation: if I had a blue patch, a green patch, and a belowness quale, then would that constitute green’s being below blue, or *vice versa*? Rather, spatial relations must be determined by a dynamic process. If we operationalise the mathematical concept of space, we see that it is the capacity for movement. That, I suggest, points to the correct model of spatial relations in the sensorium. 

My hypothesis is that the remembered pattern of movements and consequent changes of perceptual content constitutes the perception of space. Certainly, the commutative relation of movement provides a sufficient basis for the commutative relation of position. Thus if T, G, and R denote three qualitatively different sensations, and u and d denote two opposite movements (say, upward and downward) then the sequences <T,u,G,u,R> and <R,d,G,d,T> are sufficient to define the spatial relation of those qualities in the sensorium. When sensations are assembled into a topology of this kind, we refer to the resulting construction as the body image. I suggest that the memory of these volitional relations of this kind is constitutive of the apparent spatial relations: two sensations have a spatial relation by virtue of their being embedded in a body image, which is constituted by a network of remembered volitions and changes of experience.

I shall outline two examples. The first is chosen for its simplicity, which lets us introspect more clearly the principles at work; the second is chosen because it has some empirical support. First, consider the buccal cavity, that is, the interior of the mouth. Unlike the rest of the surface of the body, we do not habitually inspect the mouth either visually or digitally: although, of course, we can and occasionally do look inside the mouth with a mirror, or insert our fingers, predominantly we do not. Instead the ‘conscious mouth image’ is formed from the somatosensory experiences of the only movable piece of anatomy inside it, namely the tongue [36]. Press the tip of your tongue against the inside of your incisor teeth (sensation T), now move it upward and gain a different sensation of the gum (G), and move it upward again to the roof of the mouth (R); then reverse the movement. The memory of these two sequences, <T,u,G,u,R> and <R,d,G,d,T>, form part of the body image of the mouth. The whole mouth image, and hence all spatial relations of mouth sensations, are, I submit, constituted in this manner.

Second, suppose I am looking out from my garden table (T), and I see a green field (G) and the red sunset (R) above it. (It matters not whether this is a waking experience, or a dream or an hallucination). If I move my gaze upwards, then I find that the foveal part of the visual field, which was brown, becomes green, and after a further movement becomes red; and if I move my gaze downwards then my fovea shows green and brown again. The memory of these two sequences, <T,u,G,u,R> and <R,d,G,d,T>, and a myriad similar ones, constitute the spatial skeleton of my visual field. Saccadic movements of the eye provide a constant flux of minute movements of the eye to and fro, maintaining the visual space. On this view, if the saccades were to cease, the visual space would disintegrate. You can approximate this condition for peripheral vision simply by staring fixedly into your own eyes in the mirror for several minutes, and observing the perception disintegrate and be replaced by imaginings [37], a process related to the Troxler effect.

Barring telepathy with lower animals, we can have no first-hand data of the sensorium of other creatures. Nevertheless we can make a plausible guess, based on similarities in neuromuscular mechanisms in their respective physical avatars, that an earthworm’s sensorium is built up in the same way as the human tongue, as it navigates through its subterranean tunnels.

The precise manner in which memories of movements and perceptions are linked to form the body-image is a matter for empirical science. But those memories must, in any case, form the ground truth against which the body-image is validated, as the mind has no other data. Hence:

**Hypothesis** **4.**
*(Mindspace): Spatial relations of experientiae in the conscious mind are constituted by remembered sequences of volitions and consequent changes of experiences.*


For two experiences E_A_, E_B_, let us say they are proximal *prox*(E_A_, E_B_), if there is some sequence of volitions V_1_, V_2_, …, V_n_ and experiences F_1_, F_2_, … F_n−1_ such that E_A_, V_1_, F_1_, V_2_, F_2_, … F_n−1_, V_n_, E_B_ is a remembered compound action, and if there also exists a sequence of volitions W_1_, W_2_, …, W_n_ such that E_B_, W_1_, F_n−1_, W_2_, G_n−2_, … F_1_, W_n_, E_A_, is a remembered compound action. Obviously this is commutative, *prox*(E_A_, E_B_) iff *prox*(E_B_, E_A_). Furthermore, two experiences E_A_, E_B_ will be spatially related, *spat*(E_A_,E_B_), if either *prox*(E_A_,E_B_) or there is some sequence of experiences G_1_, G_2_, G_p_, such that *prox*(E_A_, G_1_), *prox*(G_i_,G_i+1_) for i = 1 to p − 1, and *prox*(G_p,_ E_B_). Obviously this is commutative, *spat*(E_A_, E_B_) iff *spat*(E_B_, E_A_). Note that *spat* is more general than *prox*, that is, *spat*(E_A_, E_B_) ⇏ *prox*(E_A_, E_B_) because there may be no remembered compound action all the way from E_A_ to E_B_.

If mental monism is true, then mental space must be built up from mental primitives in something *like* the above manner. The foregoing details are proposed as a speculative, but possible, account of mental space: I expect further research to discover more accurate models of *prox* and *spat*. Going forwards, we may assumed that *prox* and *spat* exist and are built up from experiences and volitions in some manner or other.

This model does rely on the notion of memory, for which no model is presented here: that is a work in progress.

The construction of mindspace concerns a level of modelling that is situated above the fundamental level that is the subject of this paper. My reason for bringing in Hypothesis 4 at all is that it entails the following corollary, which *is* required for this fundamental level of modelling.

**Corollary** **4.**
*(Mindspace not primitive): Mental space is not required as a fundamental primitive of the mental world.*


### 3.5. Hypothesis 5: Mental Individuation and Intermental Ports

Recall the distinction between a conscious mind and its subject, the latter being the featureless agent of perception and volition in the mind (see Section 2.1 above). We saw above (Corollary 2), that, without spatial location, subjects cannot be individuated, and therefore what appear to be distinct personal subjects are, in fact, numerically identical. Minds, on the other hand, can be individuated by their content, and I will now examine how this could be modelled within mental monism.

It is a matter of everyday experience that two minds M_1_ and M_2_ are mutually private but can nonetheless communicate. In physical-realism, this is straightforward to characterise and explain. Each mind M_1_ is spatially enclosed in a brain, which has afferent and efferent nerve fibres that allow communication with a shared environment, which serves as a medium for this mind to transact an exchange of information with M_2_ and other minds. In mental monism, it is not so easy: there is no space in which to contain and isolate minds, and there is no non-mental medium through which minds can communicate. Instead, we must find a way to rethink privacy and intercommunication within the sparse ontology that is dictated by the theory of mental monism. Therefore, I will first consider a model of mental privacy, using the ideas developed in the previous section, and then adapt that to model inter-mental communication.

Sensory phenomena are bounded by the sensorium (the limits of the visual field, the tactual field, and so on), but what about thoughts? As I mentioned above, any knowable thought must be identifiable by its phenomenal content. For, otherwise, how would you know what thought you had? A thought that has no phenomenal content would simply be an unconscious thought. But, is that phenomenal content sensory? Or, is there a non-sensory experience of thoughts, that is, is there a ‘cognitive phenomenology’? If so, we would need to account for not only individuation of the sensorium but also of a person’s stream of thoughts. I shall argue, on the contrary, that thoughts are indeed constructed from sensory phenomena and hence located in the body-image.

It is acknowledged that situating thoughts in the mental space of the body image might seem odd, but the alternatives are (a) to suppose that thoughts have no qualia, in which case we would never know what thoughts we have, or (b) to suppose that they have sensory qualia but outside their modal field (e.g., inwardly to hear the words of a thought, but to do so not in the inner ear), which is absurd; or (c) to suppose that thoughts are clothed in some novel qualia unlike those of any sensory modality, but (as I will argue below) such non-sensory phenomena cannot be constitutive of the thought. 

There is support in the literature for ‘restrictivist phenomenology’, the claim that all phenomenal content is sensory in nature, which is a stricter condition than mere embodiment, which is required in this paper. Carruthers and Veillet [38] have argued against cognitive phenomenology (pp. 44–45), which is the thesis that there are peculiar sensations that are constitutive of propositional thoughts; and also against there being “something it is like” to have “purely propositional—unsymbolised, imageless—thoughts” (p. 53). Prinz has argued at length for a restrictivist phenomenology of emotion [39], and against cognitive phenomenology [40]. Some authors, such as Strawson [41,42] and Smith [43] have resisted this position. Strawson’s argument [42] (p. 295), however, amounts to no more than the assertion that we engage in conscious thinking (which restrictivist phenomenology does not dispute), and does not address the claim that such acts of thinking are constructed wholly of bare sensations and memories thereof. Likewise, Smith’s starting point is the un-deconstructed experience of thinking.

It could be argued that sensory content grabs our attention and eclipses the awareness of the thought itself, but if so then blindsight [44] should allow the experience of the thought to come to the fore, in the absence visual phenomena. Not so. Type 1 Blindsight illustrates this: the blindsighter not only has no phenomenal visual experience in the affected part of the visual field, s/he has no propositional knowledge of what is viewable in that area [45] (p. 373). So, the experimental procedure is to ask the Type 1 blindsighter to *guess* the visual content, as opposed to asking the blindsighter to report knowledge of it.

The key argument against non-sensory thoughts is that they would have no operational significance. Suppose that I were to think that the angles of a triangle add up to 180°. In my mind’s eye, I visualise a particular triangle and a construction that proves the sum of the angles in 180°. In my memory, I recall times when I have carried out this construction, and times when I have used the result to find the third angle, given the other two. All I can introspect here is a web of sensory fragments. In order for a mental entity to constitute the thought in question, it must relate images of triangles and actions, such as dropping a perpendicular from a vertex and comparing angles. It must be possible to ‘cash out’ the thought in operational terms, that is sensorimotor conditions. That cash value is constitutive of the thought. Suppose, for the sake of argument, that cognitive phenomenology were true, and I were to have some peculiar, non-sensory experience whenever I have the thought that the angles of a triangle add up to 180°. Over time, I might learnt to correlate this peculiar experience with the thought. But, as it is non-sensory, it cannot form part of the operational content of the thought, and *a fortiori* it cannot constitute the thought.

At best, cognition might synaesthetically produce some peculiar non-sensory experience, although it cannot be constituted by such an experience. Could such an experience be non-embodied? First, could this experience appear to be outside the body? Suppose that every time I think that the angles of a triangle add up to 180°, I have a peculiar experience that appears to be located three inches in front of my nose. Then that would involve an extension of my body image: my body image would no longer end at my nose, but would include a new place, three inches in front of the tip of my nose. (A comparison can be made with phantom limbs, where the residual body image extends beyond the physical limb). Second, could this experience simply have no place? Could one have a non-sensory experience that is neither in the body image nor outside it, in fact nowhere? I cannot imagine such an experience. Even when I dream, my experiences are organised into a body image. In the ego dissolution of LSD trips [46], subjects characterise this in terms of an expansion of the self to include external objects, or even the whole universe, not in terms of non-embodied experiences.

This might be a failure of my imagination or introspection, so I must make it an explicit assumption that all experiences are embedded in the body-image. In principle, this assumption is slightly less strict than restrictivist phenomenology, as it asserts that all experiences are embodied, but allows the possibility that there might exist non-sensory experiences. On this basis, individuation of the mind can be accounted for in terms of the foregoing model of mindspace, as follows.

(a) *Individuation*. In mental monism, there is a universal set of experientiae that is the union of all minds’ sensoria, U_c_ = C_0_ ∪ C_1_ ∪ C_2_ ∪ … What partitions this universal set into personal subsets? What constitutes the boundaries of a personal mind? As argued above an inspection of your own sensorium right now will reveal, I believe, that all of your experientiae are situated within a personal mental space. In everyday experience, you find that you can move your attention to any area of your sensorium, thoughts, and imaginings, and can take note of that content, and act upon it, and you can initiate volitional acts to change your thoughts and imaginings and make movements of your body. You can associatively retrieve memories into attention, and lay down new memories, but your sensorium, thoughts, and memories are accessible only by you. In operational terms, therefore, it appears that your mind is closed under operations of access, and this is constitutive of mental individuation and the boundaries between minds.

This might seem odd for thoughts and emotions but, as we saw above, under the austere ontology of mental monism, all mental contents exist only as qualia, and it seems impossible to imagine them not embedded in mental space. 

As we saw above in Hypothesis 4 (Section 3.4), the mental space is woven from volitions linked to experiences, and we may therefore suppose that the boundary of the personal mind is constituted by their limit. Thus, two experientiae E_A_ and E_B_ are termed co-mental, *co-m*(E_A_, E_B_), if they belong to the same mind. Building on the concept of mental space, we can now hypothesise that the edge of mental space marks the partition of U into C_i_. Thus *co-m*(E_A_, E_B_) iff *spat*(E_A_, E_B_).

(b) *Communication.* If two minds M_1_ and M_2_ are to interact then, as we have noted above, they must carry out their inter-mental communication in an intersection of the contents of two minds, which I will refer to as a ‘port’. (The allusion is to an input/output port of a computer). P_i,j_ = C_i_ ∩ C_j_ where M_i_ = <S, C_i_, R_i_>. If a mind M_1_ changes part of its contents that is not within the port, then another mind M_2_ will not know about it. Only if M_1_ changes something within the subset that constitutes the port can M_2_ detect it.

Within the port, a change of content that is executed by one mind is deemed to be its ‘output’, and when it is detected by the other mind, it is that one’s ‘input’. In principle, a port is bidirectional: the two minds who share it could use it for input and output. In practice, ports are unidirectional, at least in complex organisms such as us. The reason for this is clear. The output from a mind—which corresponds to motor activity—is not random noise but comprises intentional acts that are ‘pre-processed’, that is planned and controlled by computational machinery sitting in the part of the mind that is outside the port. Likewise the input into a mind is ‘post-processed’, that is analysed and cognised, by other computational machinery. Therefore, input and output ports are hooked into different logic circuits. There are no ‘walls’ between one port and another, nor between a port and the rest of the mind, but the notion of ‘a port’ makes sense if its pre- or post-processing is a functionally distinct part of the mind. (A percipient mind would not normally change the content of its own input port, but doing so would be classed as dreaming or hallucinating).

Although a port can be bi-directional, its afferent and efferent faces must be separate, otherwise one mind could leak into another. If M_1_ and M_2_ are minds with a port containing experientiae X = {X_1_,..,X_N_} that are inputs to M_1_, and Y = {Y_1_,…,Y_M_} that are outputs from M_1_, then: X ∩ Y = 0, *spat*(M_1_,X) but ⌐*spat*(M_1_,Y). If Y were accessible to M_1_ then M_1_ could use it as a bridge to shift its attention into M_2_, in a kind of telepathic communication—which, if it happens at all—is not part of the normal operation of the mind. That is, M_1_ can write to Y, but cannot read from it. 

Tying together these ideas, we have:

**Hypothesis** **5.**
*(Interface): The contents of a mind are closed under operations of access, the formal relation spat forming a mental space. Communication between minds occurs through intersections, ‘ports’, between the communicating minds.*


A mind could have multiple ports with another mind, and ports of the same direction (in or out) could overlap. Although the hypothesised port model allows direct communication between personal minds, in fact almost all of the direct communication that a personal mind engages in will be with the part of the metamind that drives that mind’s avatar in the physical construct. Our sensorimotor engagement with our surroundings is mediated by the elements of the metamind that manifest as sense organs and muscles of the avatar. Likewise, the pre-conscious operations of the mind, the storage and retrieval of memories, the recognition of faces, the cognition and construction of sentences, all rely upon interaction with the metamental units that govern the brain and nervous system. The metamental ports in the personal mind are so extensive that the personal mind has little more than a central executive role to call its own. Be that as it may, our concern here is to model the basic mechanisms involved.

In the models of Hoffman [10] and Kastrup [11], the boundary of the mind is considered as a ‘Markov blanket’, a term introduced by Pearl [47]. The model proposed here is, in principle, consistent with that approach. Although the input/output ports are allowed to be anywhere within the structure of a mind, we can still regard the states of the I/O ports collectively as forming a Markov blanket, and the states of the parts of the personal mind that are outside the ports as forming the ‘internal’ states, and everything outside that mind as forming the ‘external’ states. In this model, the I/O ports are a logical boundary, but not a spatial or structural boundary. 

### 3.6. Hypothesis 6: Elementary Experientiae

The phenomenal contents of the mind exhibit a rich qualitative variety (colours, flavours, emotions), and it would be surprising if these were all primitive elements of reality. Our reductionist instinct makes us seek to explain them as composites of some more basic constituents. Mental monism, however, excludes the existence of non-phenomenal elements: if something cannot be experienced by a conscious mind then it cannot have a concrete existence (see Corollary 3, Section 2.2 above). That does not, however, mean that the basic constituents must always be noticed by the percipient. (This distinction reflects Block’s differentiation of phenomenal and access consciousness [48]).

We can find examples of this phenomenal composition in subtle sensory experiences such as the sound of orchestral music, the taste of whisky, or the colours of a painting. A trained mind can analyse a compound perception into its constituent phenomenal elements, which the untrained mind cannot notice. The unskilled palate can differentiate Laphraoig from Macallan without being able to articulate any of the component flavours involved in that difference. This does not mean the constituent sensations are not present in phenomenal consciousness: they are there, but the person does not have the wherewithal to pick them out and recognise them.

Therefore, it is at least a coherent proposal that the rich repertoire of our experientia is constituted by some small set of phenomenal primitives that are capable of being apprehended consciously but whose isolated forms are normally beyond the liminal boundary. That primitive set could even be a singleton: it is at least a coherent hypothesis, albeit surprising, that there is a single elementary experientia out of which all phenomenal contents of the mind are formed.

**Hypothesis** **6.**
*(Elementary Experientiae): All experientiae resolve into a small set of ‘elementary experientiae’, which are phenomenal primitives that are capable of being consciously experienced, and that admit of no further decomposition.*


### 3.7. Hypothesis 7: Elementary Operators

I have proposed the ‘elementary experientiae’ as the basic constituents of the mind. What are the minimal operators we need to posit for these elements?

In the ontogenesis of a mind, the new mind starts from very little, or nothing at all, and ends up with vast numbers of sensorium places. So, as a minimum, we need a creation operator; and, for symmetry, we might suppose a destruction operator. If we suppose the simplest form of experientia is a bare phenomenal existent, then the minimal operators are a correspondingly simple pair that merely create—in a mitosis-like fission, leading to two identical phenomenal cells—and merely destroy in an annihilation. Whatever the actual elementary experientiae and operators turn out to be, it is hard to imagine they would not include these two as bare minima.

Unlike the mitosis of an amoeba, the daughter cell cannot float free of the parent cell, as there is no spatial medium in which to float off. If anything did, *per impossible*, float off, it would simply be lost. We may think of the daughter cell as inhering as a feature of the parent, rather than setting off as a new existent. Does the parent cell automatically get annihilated after creating the daughter? The simpler hypothesis is that nothing happens to the parent: it simply carries on until an annihilation operator gets applied to it.

We thus have a picture of a chain of experientiae comprising successive daughter cells <m_1_, m_2_, m_3_ …, m_n_> (Figure 2a), which can grow further through the fission of the terminal cell m_n_ (Figure 2b). The interpretation of this formalism is that each ‘cell’ is a moment of experience in a mind. Although this represents a sequence in mental time, we need not assume that past experientiae cease to exist. On this view, the ‘flow’ of time is constituted by a growing chain of experientiae rather than by the changing state of a single, stateful thing.

Again for the sake of simpler hypotheses, we will suppose that there is nothing prohibiting non-terminal experientiae from creating daughters. That is, we will suppose that an experientia can undergo fission even if it is not the final cell. We might naively think of pre-final experientiae as ‘in the past’, but diachronic structure is something we are now constructing, not something already given. In this way, the chain becomes branched like a tree. There are many ways this might happen, and we will consider two plausible ones. In computing terms, the contrast is between a shallow copy (Figure 3) or a deep copy (Figure 4). Let us suppose that a nonterminal cell m_i_ has yielded a chain (<…, m_i_, m_i+1_, …, m_i+n_>, Figure 3a). If the cell m_i_ undergoes fission, then we can consider (using superscript ‘1’ for the branch) two possible models, as follows. Either a budding (<…, m_i_, <m^1^_0_>, m_i+1_, …, m_i+n+1_>, Figure 3b) which can then grow independently as (<…, m_i_,<m^1^_0_, m^1^_1_, …, m^1^_p−1_>, m_i+1_, …, m_i+n+p_>, Figure 3c). Or, alternatively, that fission of m_i_ (from Figure 4a) could be a branch duplication (<…, m_i_,<m^1^_0_, m^1^_1_, …, m^1^_n_>, m_i+1_, …, m_i+n+1_>, Figure 4b), which can then also grow independently (<…, m_i_,<m^1^_0_, m^1^_1_, …, m^1^_n+p−1_>, m_i+1_, …, m_i+n+p_>, Figure 4c).

There are two key differences between those two models. Namely, the deep copy yields a faster propagation into the future (there is no ‘deep deletion’ comparable to ‘deep copy’) and the siblings in the deep-copied branch are inherently related to the original experientiae whence they were copied. 

The weak anthropic principle can help us in selecting a plausible model. For, we know that we live in a stable and complex world, so models that do not have the resources to create such a world can be excluded from consideration. Now, whether reality involves shallow or deep copying is an open question, but as I will argue below, a universe based on shallow copying of experientiae is unlikely to have yielded a stable world, unless we bring in some rather arbitrary transition rules. A further hypothesis that enables a stable world to develop is that, in a deep copy, siblings remain neighbours. Thus suppose our initial chain (<…, m_i_, m_i+1_, …, m_i+n_>, Figure 5a) fissions at i to yield a branch, <…, m_i_,<m^1^_0_, m^1^_1_, …, m^1^_n_>, m_i+1_, …, m_i+n+1_> and again to yield a second branch <…, m_i_,<m^1^_0_, m^1^_1_, …, m^1^_n_>, <m^2^_0_, m^2^_1_, …, m^2^_n_>, m_i+1_, …, m_i+n+2_> and even k times to yield k branches, <…, m_i_,<m^1^_0_, m^1^_1_, …, m^1^_n_>, …, <m^k^_0_, m^k^_1_, …, m^k^_n_>, m_i+1_, …, m_i+n+k_>, see Figure 5.

In this model, any three successive siblings m^q−1^_j_, m^q^_j_, m^q+1^_j_ or m^q^_j−1_, m^q^_j_, m^q^_j+1_ are neighbours for j = 1 to n and q = 1 to k. For example, see the grey quintuple in Figure 5b. As we shall see below, this feature then enables us to define a cellular automaton over the experientiae, which otherwise would not be possible.

A third speculative hypothesis that is motivated by the weak anthropic principle concerns the transition function. In line with Hypothesis 1 (Naturalism), we may make two assumptions about the operation of the fission operator. First, that the operator behaves orthonormally, obeying some laws that we must discover by scientific inquiry—which might be partly or wholly stochastic. Second, that the fission operation is driven by its neighbourhood of other experientiae. That is, at each point in time, whether a given experientia stays unchanged, or annihilates, or divides into a daughter, this decision is governed by a transition rule that may be a mix of determinism and randomness, and is based on the states of its neighbours.

**Hypothesis** **7.**
*(Operators): There exist at least two operators acting upon individual experientiae: creation and annihilation. The creation operator acts upon an experientia to make an identical copy of the experientia and all its descendants. Each descendent will become a neighbour of its counterpart in the created chain of new descendants. The annihilation operator permanently destroys an experientia. The application of these operators at time t is a function of the pattern of existing neighbours at t − 1.*


Which experientiae count as the neighbours of a given experientia is an open question. We have a wide range of possibilities. One such possibility is that only the siblings are effective as neighbours. If there are multiple deep copies made in rectilinear manner, so that m_i_ fissions to m^1^_i_, which repeats the exercise to become m^p^_i_ to m^r^_i_, and if each m^q^_i_ also fissions into an orthogonal chain m^q,1^_i_, m^q,2^_i_,… then we have a series of infinitely extensible two-dimensional grid-like layers. Each ‘cell’ in the plane functions as a unit automaton, which can be occupied by an existing experientia or be empty, and can change between those two conditions in accordance with transition rules based on the neighbours. In this specific case, we have a classical two-dimensional cellular automaton [49]. 

It is a standard result that some binary-state cellular automata are capable of functioning as a universal Turing machine, and could therefore be capable of computing any computable function [50]. As this planar cellular computer could be produced by initiating deep copying from any starting point, we have a hierarchical computational structure that constitutes an object-oriented architecture. In fact, it has been shown that one-dimensional cellular automata can implement universal computation, albeit with a more than binary state space. This was conjectured by Wolfram in 1985, and a proof by Cook was presented in 1998 and published in 2004 [50].

Of course, the suppositions involved in producing this classical cellular automaton are deliberately contrived to yield that result. Quite different facts may obtain in nature. The point of the exercise, however, is to offer an existence proof: we have shown that this model of the basic units of conscious experience encompasses at least one instantiation of a universal computer. The model does, however, provide a vastly richer space of possible instantiations, some others of which might enable universal computation. 

### 3.8. Hypothesis 8: Stochastic Transitions

The actual attributes of elementary experientiae, and of their transition functions, are completely unknown at this stage. I have argued above that the operators on elementary experientiae will include creation and annihilation. We do not yet have tools to access the elementary mentations in order to study them scientifically. Our twin starting points will be: (a) computer simulations of possible permutations of the basic model; and (b) studying the logical structure of the neural correlates of consciousness (which we do not know for sure yet).

One of the components of the model to be settled is the transition function. In conventional studies of cellular automata, we define the transition function *ab initio*. For example, in John Conway’s widely studied *Game of Life* [49], the rule is that a unit automaton is created when an empty cell is surrounded by three other unit automata, but is annihilated if surrounded by zero or four. There is, however, a question of inelegance here. Why would the universe, at its elementary level, be governed by a rule as arbitrary as Conway’s? From a subjective point of view, it would seem more elegant to avoid prescribing any particular deterministic transition. Instead, I want to suggest a wholly stochastic mode of operation, for example, that an experientia has a fixed probability of being annihilated (say 50%) at the next step, and if it persists then it has a fixed probability (say 50%) of breeding through deep copying of descendants). At some level of annihilation probability (not necessarily 50%), the statistical expectation is that the system would survive and grow, as opposed to fizzling out. Whether a purely stochastic transition rule would actually yield interesting behaviours, at least one capable of universal computation and self-reproduction, is unknown. We might have to consider such exotica as retroactive transitions, or transition rules that can change in time. At present, the modelling of cellular automata of conscious experientiae is virgin territory.

**Hypothesis** **8.**
*(Stochastic transitions): The transitions of elementary experientiae are wholly stochastic.*


I would emphasise that I have no philosophical or empirical grounds for this particular hypothesis: I put it forward only for reasons of internal elegance. 

Unlike any physicalist or dualist model, the mentalist model allows conscious mechanisms to exist without any physical correlate, except for the boundary condition of compliance at the point of neural correlation. In other words, the model in principle allows disembodied thought processes.

## 4. Discussion

As mental monism is a minority position, there are correspondingly few writers who have addressed the modelling of the conscious mind from that position. I will briefly compare and contrast the present model with the proposals of Donald Hoffman and Bernardo Kastrup.

### 4.1. Donald Hoffman

(a) *Philosophy.* Chalmers [3] classified Hoffman as a ‘macro-idealist’, but Hoffman himself shuns this term. Hoffman has previously asserted [51] that “MUI [Multimode User Interface] theory is not idealism. It does not claim that all that exists are conscious perceptions”, but more recently he has aligned himself [10] with the foremost Western idealist: “George Berkeley clearly summarised some of the key ideas [of conscious realism and the interface theory of perception]: ‘For as to what is said of the absolute existence of unthinking things without any relation to their being perceived, that seems perfectly unintelligible. Their esse is percipi, nor is it possible they should have any existence out of the minds or thinking things which perceive them’”. The apparent contradiction was explained by Hoffman in a recent interview [52] (position 38:15): “I’ve noticed among my colleagues, if I use the term ‘idealism’, that it was an anti-science, anti-realist point of view, which is why I then decided not to use the term ‘idealism’ as there are too many things that get packed into the term, so I’ve decided to use ‘conscious realism’”. Thus Hoffman’s philosophy of ‘conscious realism’ is what Lloyd [5] calls “the philosophy that dare not speak its name”, namely ‘idealism’, or ‘mental monism’.

(b) *Architecture.* Hoffman [10,53] has proposed a mental monist theory in which the physical construct arises from interactions among a community of equipotent ‘conscious agents’. At first sight, this seems contrary to the model proposed here, in which a single entity (the metamind) imposes the construct upon the community of personal minds. In fact, Hoffman allows any interacting conscious agents to combine to form larger conscious agents, so an army of cooperating conscious agents could be combined to form a single agent. On the other hand, ‘metamind’ is used here as a collective term for the conscious mind(s) that govern experiences outside personal control. The difference is therefore one of emphasis, not substance. 

Hoffman models the states of a conscious agent as a ‘Markov kernel’—a series of states in which each state change depends on only the preceding state and inputs into the agent. The agent is partitioned into an interior and a ‘Markov blanket’, where only the states of the blanket interact with the outside world. As I mentioned earlier, the I/O ports of a personal mind collectively form a Markov blanket, although the boundary of the personal mind is highly distributed. For, the mind performs a myriad tasks that are not under direct conscious, voluntary control. To give two mundane examples: as you read these words, a pre-conscious mechanism recognises the pattern of black-and-white marks, and delivers the meaning to your personal consciousness; and as you move the cursor across the screen, your intentions are post-consciously converted into muscular contractions in your mouse-holding hand. In the model proposed above, these functions must be performed by the metamind, which controls any actions not under volitional control by a personal mind. In fact, almost all mental activity involves a degree of sub-conscious functioning. Therefore, almost all data paths through the personal mind are routed through multiple ports with the metamind. This is closely analogous to computer software: a compiled Fortran program will execute segments on its own, but will often need to invoke system routines for input and output to other devices, for communicating with other process, for allocating and deallocating memory, and for performing common but nontrivial computations such as statistical formula. Nevertheless, formally we can group these extensive ports into a single Markov blanket. 

(c) *Model.* Hoffman’s general aim coincides with the one sought in this paper. Hoffman and Prakesh [54] state: “[S]pace-time and three-dimensional objects have no causal powers and do not exist unperceived. Therefore, we need a fundamentally new foundation from which to construct a theory of objects. Here we explore the possibility that consciousness is that new foundation, and seek a mathematically precise theory”. 

The term MUI (mental user interface) was introduced by Lloyd [19] to denote a view of the physical construct that enables a conscious mind to have a single structure with sensory content and volitional handles, through which it can interact with whatever ‘external world’ exists outside the personal mind. The analogy with a GUI (graphical user interface) is obvious. Hoffman [51] independently reintroduced the term MUI (multimodal user interface) for the same idea, and his theory covers both the mechanism of the MUI and the evolutionary development of it.

Fields, Hoffman, Prakash, and Singh [55] offer the fullest account of Hoffman’s model so far, but it is outside the present paper’s scope to review this substantial work fully, beyond noting points of contact with the model proposed here.

Hoffman and his collaborators [54,55] explicitly compare the status of the ‘conscious agent’ model with that of Turing’s ‘machine’. Turing [56] described his automaton as an abstraction, even though he used the metaphor of a machine with a tape running through it and could scan and write symbols in discrete squares on the tape. He offered no model of how the ‘tape’ was moved, or how the ‘read/write’ head functioned, nonetheless the model proved to be of immense value. In like manner, Hoffman offers no model of the internal structure of the state of a conscious agent, or the internal dynamics of perceptions, decisions, or actions. In contrast, the model presented here specifically address those questions, while largely eschewing discussion of the higher level dynamics. The models are, in this respect, complementary.

The conscious agent is a generalisation and formalisation of our everyday idea of a conscious mind: it has a space of possible states, a space of possible perceptions, and a space of possible actions. These are abstract ‘spaces’ (which Hoffman [10] (p. 189) called ‘menus’ of possible experiences etc.), and unrelated to either physical space or the mental space considered above. The scale of a conscious agent is not defined, although Hoffman has suggested a microphysical role for them [10] (p. 204), but they can be aggregated, allowing for complex conscious agents such as people.

Hoffman [10] uses the same term for the conscious agent and for its mathematical model. On p 188, he writes that a conscious agent ‘perceives, decides and acts’, which are actions ascribable to conscious minds. On the same page, however, he says that a conscious agent contains measurable sets, but a measurable set is a 2-tuple <S, M> where M is a subset of the power set of S, S ⊆ P (S), which means that a conscious agent is in the ontological class of mathematical abstractions. Even if the elements of S are ingredients of a real-world mind, the elements of M are artefacts of the description of S. Hoffman confirms this further down the page: “The definition of a conscious agent is just math… the mathematical model of conscious agents is not, and cannot create, consciousness”. Here, let us disambiguate his nomenclature by using ‘actual agent’ for the actual conscious agent and ‘model agent’ for the mathematical model of the actual agent. This is not a semantic haggle but a substantive problem, as will become apparent. The model agent contains a measure M and the modeler is at liberty to define M any useful way. In contrast, the actual agent has no measure: there is no fact of the matter whether the actual mental elements have any particular set of subsets. (Otherwise, Hoffman would have to propose that sets are actual constituents of reality, alongside experiences and volitions. That, however, would be a move away from mental monism, which seems to be a fundamental tenet of Hoffman’s work).

Hoffman needs to introduce measurable spaces into his theory only because he allows the contents of the mind to be continuous. For, the power set of a continuous space includes many ‘perverse’ non-measurable sets, but probability theory applies only to measurable spaces, and he therefore has to exclude the ‘perverse’ sets. In a finite, discrete space, however, the power set of the space is adequate as a basis for applying probability. Hence, in the present paper, the conclusion of discreteness allows us the economy of not bringing in measurable spaces. As I argued above (Result 1, Section 2.2), the contents of consciousness can be assumed to be discrete. Therefore, Hoffman’s use of measurable spaces in order to allow for continuous sensoria seems unnecessarily heavy-handed.

Hoffman then asserts that “every aspect of consciousness can be modelled by conscious agents”, but he excepts qualia, regarding which he already asserted on p. 44, “I will avoid this term because it often triggers debates about its precise definition. I will instead refer to conscious experiences”. Correspondingly, the modelling stops short of experientiae: “There is a bottom to the hierarchy of conscious agents. At the bottom reside the most elementary agents—‘one-bit’ agents—having just two experiences and two actions. The dynamics of a one-bit agent, and of interactions between two such agents, can be analysed completely”.

But it is not just qualia that his model does not address. Non-spatiality, and its ramifications, are not explicitly handled. Any formalism is just a formalism: Hoffman’s model, and the model proposed here, can both be applied to non-conscious information processors as well as conscious minds. In the model proposed here, however, the hypotheses are framed so as to derive specific formal consequences from the philosophically argued features of consciousness such as its non-spatiality. Although Hoffman starts likewise from mental monism (his ‘conscious realism’), his model of conscious agents is not explicitly derived from features of consciousness as such.

The full definition of Hoffman’s model agent is: “A conscious agent, C, is a seven tuple C = (X, G, W, P, D, A, T), where X, G, and W are measurable spaces, P: W × X → X, D: X × G → G and A: G × W → W are Markovian kernels, and T is a totally ordered set” [10] (p. 203). The intention is that X represents the set of possible states of the actual conscious agent (without explicit regard to how each state is constituted by sensations and thoughts), G the actions, and W the outside world. A ‘Markovian kernel’, more commonly called a Markov chain, is a stochastic transition function, so P represents perception of the world, D represents decisions, and G represents actions. T is discrete time. Hoffman’s hypothesis is that the world, W, consists of conscious agents. Activity within and between model agents is represented by the Markovian kernels, but Hoffman does not model those functions: there is no explanation of why the kernels are such and thus, or what mechanism makes them tick. Such questions are relegated to a lower explanatory level.

Hoffman and his collaborators [55] derive a number of results concerning cognition and memory, but these operate at a higher level of description than the model proposed here, and will not be examined. They also propose that physical space could be implemented in model conscious agents. The treatment of this, however, appears to be incomplete. Lloyd [5] (Sections 4.1 and 4.2) showed that positing a pre-spatial mental world creates a risk of superluminal communication, which is impossible for fundamental reasons. Lloyd offers one possible resolution of this conundrum. But Hoffman *et alia* do not explicitly address the problem. In this connection, we should also note that this method does not, and cannot, yield a mental space (as is discussed under Hypothesis 4), because a mental space is concrete, not abstract. For example, if you see a blue patch above a green patch in your visual field, then that is a concrete fact and cannot depend on the modeller’s imposition of a set-theoretic space upon conscious elements. Whatever the mechanism that creates mental space, it must be a concrete mechanism, even though of course it may be described by a formalism (as indeed I have attempted above). This is why is matters whether the ‘conscious agent’ is a mathematical abstraction or an actual conscious entity: we are at liberty to define abstract spatial relations on the states of a model agent, but what needs explaining is the perception of concrete mental spatial relations between experiences. 

In conclusion, Hoffman’s model and the one presented here are applicable to different levels of description. They appear to be complementary: the model proposed here neither entails nor excludes Hoffman’s higher-order model, although there appears to be a difficulty in reconciling Hoffman’s space construction with the risk on superluminal communication between pre-spatial conscious agents.

### 4.2. Bernardo Kastrup

Kastrup [11,57,58] explicitly advocates an idealist philosophical position, and acknowledges the need for idealism ultimately to re-ground the whole of physics [11] (pp. 101, 120). Kastrup often describes his work as ‘rigorous’ (e.g., about twenty times in [11]), and he is widely cited in the literature, and he is therefore included in this discussion.

Kastrup [11] (Chapter 5) lists a number of “Basic Facts” that inform his theoretical development. The fifth one (p. 62), which he says is “self-evident” is, “Fact 5. Irrespective of the ontological status of what we call ‘a person’, there is that which experiences (TWE)”. The TWE is a concept that he uses throughout the book, and explains it thus: “I am not necessarily making an ontological distinction between experience and experiencer here; in fact, soon I will claim precisely that there is not such a distinction. I am simply recognising that experience necessarily entails a subjective field of potential or actualised qualities. TWE is this field”, and (p. 63) “experience is a pattern of excitation of TWE”. Later (p. 242), he expands on how this field is supposed to yield the contents of conscious experience: “This disposition to self-excitation is inherent to universal consciousness itself and entails certain natural modes of excitation, much like a drumhead vibrates according to certain harmonics and not others. It is self-excitation that allows the structure and complexity of manifest nature to arise from the undifferentiated ground of universal consciousness”, and he compares the mind to Chladni patterns [59].

The first basic difficulty that this position faces is that the existence of the TWE as a “field” presupposes a space in which the field is extended (a field being defined as a function that maps a value codomain onto a space), which contradicts the idealist premise that consciousness is pre-spatial: space has to be constructed out of units of consciousness, not the other way around. Second, excitations of this field, which apparently have different amplitudes in different positions in this pre-conscious space, imply an energy that drives the excited waveforms. But that takes us further from mental monism, as it adds another fundamental constituent of reality besides consciousness. Third, we are brought back to the explanatory gap of Chalmers’ *desiderata*: even if we suppose there is such a TWE field as the ground of reality, undergoing Chladni-like vibrations, why would that have any consciousness associated with it? To be sure, consciousness can be formally modelled, but it has to start with the consciousness, not the formalism.

The second strand of Kastrup’s model is his partitioning of the universal TWE into personal minds [60]. He wrote, “We and other living organisms are dissociated alters of this universal mind, akin to the multiple disjoint personalities of a person with dissociative identity disorder [DID]” [11] (p. 54). That the universal set of conscious experiences must somehow be partitioned into personal minds is an immediate consequence of any version of mental monism. Comparing this with dissociative identity disorder, in which a personal mind is partitioned into sub-personalities, does not illuminate the ontogenesis of personal minds but does add metaphorical baggage, as DID is a pathological condition usually produced by psychological trauma. Kastrup does not offer a formal model for how the partition is formed and sustained. He did write, “The boundary of an alter is thus *akin* to a Markov Blanket… For this reason, and inspired by Friston’s model…, I shall represent the interaction of an alter with its surrounding mental environment” (p. 113) [my emphasis]; he subsequently asserted that the alter boundary “is” (not “is akin to”) a Markov blanket between alters but did not offer a formal model on which to base this.

In conclusion, Kastrup offers several metaphors but only hints at a formal model.

### 4.3. Cytoskeletal Cellular Automata

Penrose and Hameroff [17] have (a) theorised that moments of conscious experience are identical with, or supervene on, the orchestrated objective collapse of tiny quantum superpositions inside biological cells; they have also (b) hypothesised that the physical correlate of consciousness in the brain is the microtubule. Regarding (a), neither Penrose nor Hameroff are idealists, and the philosophical theory of mental monism excludes the Penrose-Hameroff philosophy of identifying qualia with, or supervening them on, the objective collapse of quantum superpositions in the microtubule. Nevertheless, regarding (b) their arguments are attractive for regarding the microtubule as the locus, in a person’s avatar, of the physical correlate of consciousness.

At this point in the discussion, it might be useful to rehearse the reason that an idealist philosophy even requires a neural correlate of consciousness, since, after all, the brain does not actually exist according to mental monism. Lloyd [5] (Section 3.2) discusses the ‘physical construct’ in detail, but this can be recapped here. Since we know that the community of minds M_1_, M_2_, … communicate with each other, some protocol must exist whereby the actions of M_1_ can be made known to M_2_
*et alia*. In principle, this could have been implemented pairwise, so that M_1_ and M_2_ communicate with each other in their own protocol that is independent of the protocol between M_1_ and M_3_, say. Nature has not done this, but has rather evolved a system in which the metamind, M_0_, maintains a common protocol in the form of the physical construct. Minds use the physical construct like a bulletin board: M_1_ can post a message to be read by M_2_ and M_3_, and can read messages posted by other minds. The specific protocol for a mind to post a message involves a data structure called the ‘avatar’, which is an object in the form of a human body inside the construct. Each avatar can use physical processes within the virtual world to communicate with other avatars and hence other minds. The specific substructures within the avatar that the mind’s input-output ports map on to are the physical correlates of consciousness—tiny structures inside the central nervous system that have a degree of physical non-determinism. We do not yet know what those structures are, but we know they must exist in some form, and the Penrose-Hameroff hypothesis is that they are microtubules, although other cytoskeletal structures are also plausible.

Human beings faced a logically similar problem of communication, and devised electronic bulletin boards in 1978, which evolved into the world-wide web in 1989, where people communicate now through avatars in chat rooms. This metaphor might make it sound like the physical construct might have been designed, as the internet was, but we can safely assume that the physical construct arose by blind chance and natural selection. 

As recounted by Penrose and Hameroff [17], there are two key observations that warrant investigating cytoskeletal structures as correlates of consciousness: first (p. 42), neurons that are involved in conscious brain processes apparently deviate from the standard Hodgkin–Huxley model of neuronal signal propagation, suggesting that an intracellular mechanism may also be at work; second (p. 44), unicellular organisms exhibit complex behaviour without the benefit of a neural network, but apparently based on cytoskeletal information processing. Cytoskeletal strands such as microtubules possess a lattice structure of tubulin proteins, each capable of rapid changes of conformation or dipole state, which may involve nondeterministic quantum-mechanical superpositions. This warrants investigating the ability of cytoskeletal structures to operate as one-or two-dimensional cellular automata, which might also be able to operate as physical correlates of consciousness.

This is of interest here because the model proposed above suggests that, in the mental monist theory, reality consists of units of conscious experience that might operate together as cellular automata. If this is correct, then we would expect the physical correlates of consciousness to be mechanisms that can interact with mental cellular automata, and cytoskeletal cellular automata are therefore candidates worth considering.

It has been known since the 1980s the cytoskeletons could, in principle, sustain cellular automata [61]. Recent work [62] demonstrates hybrid quantum-von Neumann cellular automata. So far, this work has been based on proof-of-concept computer simulations. If this concept were to prove successful, then subsequent stages would involve experimentally establishing in vitro computation in actual cytoskeletal fibres, then discovering natural uses of such computations in vivo, and finally determining the connection with consciousness. This avenue of research is therefore at an early stage but it has an indicative road map.

How might a conscious automaton (such as I suggested speculatively above) interface with a cytoskeletal automata? Needless to say, we do not know yet, but the following is an indicative illustration. Suppose that a conscious automaton M(t) = <S, C(t), R(t)> is organised as to deliver outputs to port C_P_ ⊂ C, comprising a two-dimensional array of experientiae C_P_[i, j] for i = 1 to 2, j = 1 to N, where C_P_[1, *] are all present, while C_P_[2, *] may be present or absent. Suppose a segment of a cytoskeletal automata (implemented, like everything else as a virtual object in the metamind) has tubulin proteins T[j] for j = 1 to N, each in superposition of two states S_A_ and S_B_ (which could be conformational or dipolar). Then, in the next time step, each of T[i] collapses so that T[i] goes to S_A_ if C_P_[2, j] is present, or S_B_ if C_P_[2, j] is absent. Thus, each experientia in the output port controls the collapse of one tubulin.

This is, obviously, a contrived example, but it serves to indicate the *kind* of interface that might work. It is in opposition to the philosophical component of the Penrose-Hameroff theory, as it implies that the conscious event causes the objective collapse of the quantum superposition (which, according to mental monism, occurs within the virtual brain in the physical construct), whereas in the Penrose-Hameroff philosophy, the moment of conscious experience is an epiphenomenal product of the objective collapse (which Penrose says is triggered by gravity, and not by the conscious mind). In this respect, the model presented here is aligned with Stapp [63,64] rather than Penrose and Hameroff. A discussion of the quantum measurement problem is outside the scope of this paper, but we can note that consciousness’s collapsing of superpositions is an unavoidable result of mental monism, whereas in Stapp’s theory (which is not idealist) consciousness is introduced as a contrivance to resolve the measurement problem.

### 4.4. Digital Physics

‘Digital physics’ refers to a class of theories in which the physical world is the result of a computation of some sort. Strictly speaking, the model proposed here is an antecedent to a possible digital physics, but such a physics would stand aside from other theories that are normally considered to be in this class, in two regards:It would be derived from mental monism, whereas other digital physics theories either have information as the ontological primitive, or have no clear statement of fundamental ontology.It denies the real existence of the physical universe, whereas digital physics theories generally regard the physical world as being real, albeit derived from a computational system.

In this section, I will briefly consider what points of contact, if any, that a few of these theories may have with the mental monist model proposed above.

Zuse [65] seems to have been the first to propose that the physical world is computed, and suggested a cellular automaton as the computer. Apart from the basic physics problems of reconciling relativistic spacetime with a universal grid, and quantum mechanical indeterminacy with a mechanistic conception of computation, Zuse’s monograph does not elaborate on his motivation for this work, or the ontological nature of the proposed cellular automaton: if the physical world is but a computation, then what is the computer made of? And Zuse does not discuss consciousness.

Wheeler proposed ‘it from bit’ in the final phase of his developing ideas on the foundations of physics. He was, however, emphatically opposed to the notion that consciousness *per se* had anything to do with quantum measurements [66], and certainly did not regard ‘it from bit’ as being related to idealism. Whatever merits ’it from bit’ may have as a pragmatic approach to physics, its lack of a fundamental ontology was clearly admitted by Wheeler: “Physics gives rise to observer-participancy; observer-participancy gives rise to information; and information gives rise to physics” [67] (p. 314). It does not count as an actual theory of the reality, but only as a promissory methodological placeholder that might one day to lead to a fundamental theory. Wheeler embraces the unfinished nature of the ‘it from bit’ notion. Observer participancy is a key element of ‘it from bit’, and yet he wrote, “What is ‘observership’? It is too early to answer. [..] The main point here is to have a word that is not defined and never will be defined until that day when one sees much more clearly [..] how the observations of all the participators, past, present and future, join together to define what we call ‘reality’” [68] (p. 26). Despite this open-endedness, Wheeler explicitly omits consciousness from his scheme. Consequently, ‘it from bit’ does not point toward any formal model of consciousness.

Tegmark [69] (p. 254) proposes two hypotheses: “*External Reality Hypothesis (ERH):* There exists an external physical reality completely independent of us humans. *Mathematical Universe Hypothesis (MUH)*: Our external physical reality is a mathematical structure”, and concludes (p. 260) that, “So the bottom line is that if you believe in an external reality independent of humans, then you must also believe that our physical reality is a mathematical structure”. This coincides with the conclusion drawn by Lloyd [5], (Section 2.3). From this, Tegmark infers that the ultimate reality is mathematical, while Lloyd [5] and Kastrup [70] draw the opposite conclusion that the physical world is a convenient fiction, while ultimate reality is mental. Tegmark is, like Wheeler, dismissive of consciousness’s involvement in quantum measurement: He then disregards it *tout suite* from physics: “understanding the detailed nature of human consciousness is […] not necessary for a fundamental theory of physics” (239). Insofar as Tegmark has any theory of consciousness it is an egregiously naïve one: “the only assumption I’m making here is that your subjective consciousness results in some way from the remarkably complicated motions of the particles that make up your brain” (p. 210), and “I think that consciousness is the way information feels when being processed in certain complex ways” (p. 290). Again, Tegmark’s book does not point toward any formal model of consciousness.

Bostrom [71] begins from a similarly naïve notion of consciousness: “Provided a system implements the right sort of computational structures and processes, it can be associated with conscious experiences”. From this premise, he constructs a science fantasy in which people ‘upload’ their conscious minds into digital computers, where they live in a simulated world, and in which new-borns spend their lives in the simulation. To this fantasy he applies the following statistical inference: when this technology comes about, it will allow numbers of conscious minds to exist in simulation orders of magnitude in excess of the number of humans who lived in the years before consciousness uploading was invented; therefore, any given conscious life (such as ours) are much more likely to be simulated than native. As argued by many (e.g., Foster [8], Lloyd [5]), the functionalist account of consciousness is not tenable, and Bostrom does not develop any alternative model of consciousness or of how a mind could be ‘uploaded’ into a physical computer.

A form of digital physics that is more promising from the perspective of mental monism is that of Wolfram [72]. He proposes cellular automata as a fundamental mode of organisation of not just physics but also of derivative complicated systems, such as those found in biology. He writes, “In the end it will turn out that every detail of our universe does indeed follow rules that can be represented by a very simple program—and that everything we see will ultimately emerge just from running this program” (p. 554). Like others in the field of digital physics, Wolfram proposes no substrate: physics is supposed to come into being through the operation of the cellular automata. Yet, if there is no substrate then there is no automaton, and the entire edifice cannot be founded. Nevertheless, if the mental monist theory that was outlined above is true, and experientiae function as cellular automata, then there would have to be at least two points of contact between the mental world and the physical construct, which would require interfacing with mental cellular automaton: first, the metamind’s control of basic physical phenomena; second, the personal mind’s input/output. If Wolfram is right, then his cellular automata would be found both in fundamental physics and in any higher-level structures, and it could therefore offer the appropriate mode of interface for both the metamind and the personal mind.

### 4.5. Evolution

Evolution is central to Hoffman’s general approach. Kastrup also touches on it. Opponents of straw-man Berkeleianism suppose that it rests on intelligent design, which as we noted above is explanatorily impotent. It does, however, highlight that mental monism owes us an explanation of how the world as we know it comes to be. I suggest that such an explanation might be formulated within the automata-theoretic approach sketched out here.

Consider the primordial soup of phenomenal elements before the formation of the physical construct. A structure of experientiae that acquires a capability for persistence will persist while others that have a purely ephemeral formation will disappear. Persistence means the persistence of a volitional structure. Suppose that, within that structure, a sub-structure chances to arise that cannot only persist but also reproduce itself. By Darwinian pressure, it will prevail over the rest of the primordial soup. If such as structure can build a physical construct with the fundamental elements of physics such as quantum mechanics, then the bootstrap is complete: the rest is history.

## 5. Conclusions

Despite the solid philosophical arguments that have been advanced for mental monism [5,21], this philosophical doctrine often encounters strong opposition. In part this seems to be due to the absence of any workable model for the elementary structure and dynamics of conscious experience, and the superstructure that would need to be built upon it if mental monism is true—namely the personal conscious mind and the manifold of observed regularity that has been modelled with physical laws over the past three centuries. If mental monism is true then it must ultimately re-ground physics in consciousness. This is a colossal project, and the present paper addresses only a fragment of the foundations of that project.

The arguments given above show that mental monism imposes severely parsimonious limitations on the possible elements of a model of consciousness. These, in turn seem to entail constraints on any reasonable model. First, to recap, the following premises are taken from Lloyd [5] and the arguments for them have not been repeated in the present paper:Central premise (mental monism), that reality consists only of conscious minds. Two Corollaries: First, that consciousness is not situated in physical spacetime; second, that there is at most one subject.

The implications of mental monism that are proposed in the present paper are as follows. First:Third Corollary of mental monism, all concrete entities that form part of the model of consciousness must be capable of being experienced.First result from mental monism, that the conscious mind is a discrete system, not a continuous one.

Second: There are then five core Hypotheses that seem to be required for modelling mental monism, although no proof is given here, followed by three speculative Hypotheses that seem plausible within mental monism but are by no means certain.

Core Hypotheses are as follows. First Hypothesis, that the elements of consciousness operate under mediated causation, not spooky unmediated action. Second Hypothesis, the elements of consciousness exhibit a nondeterministic volition. Third Hypothesis, there is a single elementary type of unit of consciousness that has both the character of experience and the character of volition. Fourth Hypothesis, that mental space is constructed from remembered patterns of volitions and experiences. Fifth Hypothesis, that mental individuation is constructed by closure under operations of access within mental space. (Hence, neither mental space nor mental individuation are required as primitives in a parsimonious model).Speculative Hypotheses are as follows. Sixth Hypothesis, that there is small set, possibly a singleton, of elementary types of experientiae, out of which the familiar qualia are constructed. Seventh Hypothesis, that there are two fundamental operators applying to elements of consciousness, namely creation and annihilation, and that the creation operator performs a deep copy on the tree of descendants of a unit of consciousness. Eighth Hypothesis, that the transition function of experientiae is stochastic.

I have argued above that these Premises and Hypotheses allow us to formulate a class of formal models of the conscious mind, some of which can sustain universal computation in the form of cellular automata.

The arguments above indicate a tentative basic model, and suggest a two-pronged research programme: On the one hand computer simulations of the relevant kind of cellular automata; on the other hand investigation of physical correlates of consciousness that exhibit behaviour akin to cellular automata. The goal is, obviously, a long way off.

## Figures and Tables

**Figure 1 entropy-22-00698-f001:**
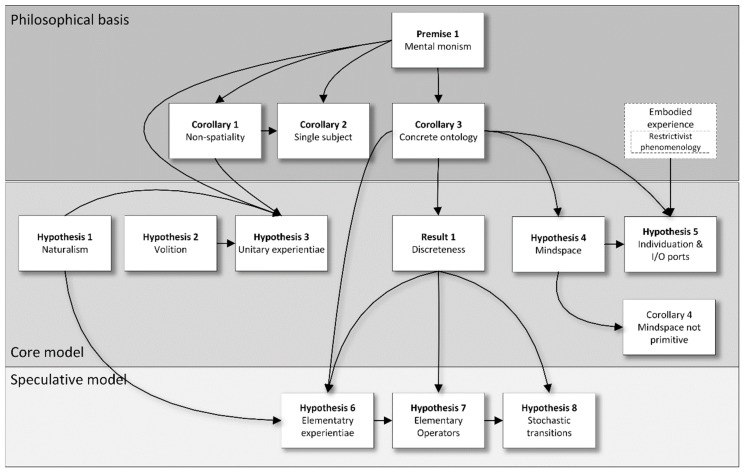
Structure of the paper.

**Figure 2 entropy-22-00698-f002:**
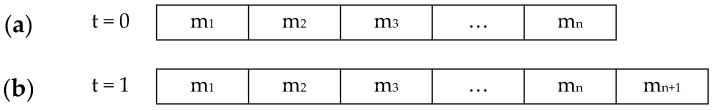
(**a**) A series of successive experientiae. (**b**) After the terminal experientia has fissioned.

**Figure 3 entropy-22-00698-f003:**
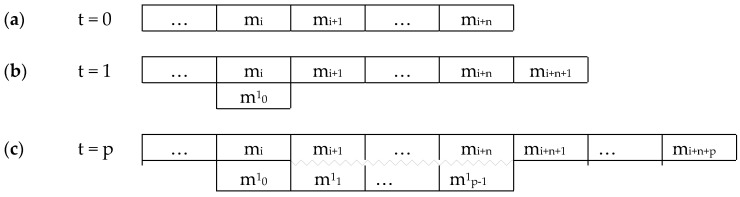
(**a**) Initial state; (**b**) one experientia buds; (**c**) the new line grows for p moments.

**Figure 4 entropy-22-00698-f004:**
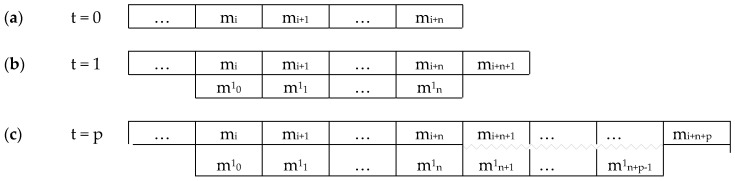
(**a**) Initial state; (**b**) one experientia deep-copies; (**c**) the new line grows for p moments.

**Figure 5 entropy-22-00698-f005:**
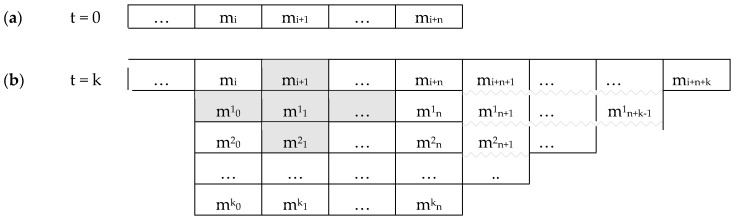
(**a**) Initial state; (**b**) after k deep-copies from m_i._

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
