# Peer review of "Modelling Consciousness within Mental Monism: An Automata-Theoretic Approach"

_entropy, 2020, doi:10.3390/e22060698_

Round 1

Reviewer 1 Report

This is a well-written paper that seems suitable for this Special Issue.

There are a few issues that need to be addressed:

1. Competing "mental monist" approaches, at least the ones discussed later in the paper, need to be mentioned and cited in the Introduction.

2. The model here depends, like Tononi's work (and most philosophical models), rather too much (in my opinion) on intuitions or assumptions about *human* experience. The author might consider how the model would deal with the experiential world of a tree or a bacterium, which might be expected to have "mental spaces" with very differe structures than ours. 

3. A. figure illustrating the distinction between shallow and deep copying would be very helpful (ln. 497-504).  The notation used in hard to follow; a visual image would clarify it.

4.

Section 3.1 on Hoffman seems not quite fair. Hoffman is fairly clear in distinguishing the "map from the territory" although admittedly abuses the "CA" notation. But Hoffman's approach is not as ontologically sparse as the current one. Hoffman's (model) CAs include, as outlined in ln. 629-637, explicit representations of the kinds of interactions that are rejected under "Result 2" of the present work. Hence Hoffman's ontology is not an ontology of experiences partitioned somehow into "minds" as proposed here, but rather an ontology of interacting entities that "have" experiences and "execute" actions. The CA theory is not, moreover, intended to represent only *human* minds; hence it makes no specific assumptioons about "mental spaces" based on human experience, but rather leaves the structure of the space of experiences entirely open (hence using measurability to allow for the possibility of continuous spaces, although there is no evidence for this as noted). This is not an oversight, but is intentional.

A somewhat similar concern applies to the (much briefer) discussion of Kastrup. It's far less clear what Kastrup's ontology is, but it may not be as restrictive as the one employed here.

5. The discussion of Penrose and Hameroff also needs work. They are clearly not mental monists, as they assume an underlying spacetime. Elementary qualia are identified with "objective reductions" (collapses) of superpositions of curvature elements in this assumed spacetime. The assumption of Orch-OR is that such reductions are "orchestrated' when they occur in microtubules (but not "orchestrated" when they occur in other macromolecules). 

6. A more general issue here is to ask why, in a mental monist ontology in which "the body" is an avatar, there should be any "NCCs" or "avatar correlates" of conscious experiences (ln. 679-681). Any such supposition seems entirely unmotivated, whether in this theory or any of its competitors.

7. There are a few wording issues to correct.  More importantly, there are mismatches between reference numbers in the text and in the reference list that need fixing.

Reviewer 2 Report

I believe this is a good and important paper that should be published. The idea that what we call perception is the influence of meta-mental volition from the meta-mind is particularly interesting, especially in light of the empirical data about modes of dissociation in which a subject loses the sense of ownership of its own mental contents (e.g. remembering one's own memories as if they were another person's memories). Our perceptions could thus be imaginations of which we've lost the sense of ownership and volitional control, thus believing they come from outside.

The critical comments below aim at improving the paper, not discouraging the editors from accepting it.

-- On bootstrapping: under mental monism, physicality is mental (for there's nothing but mentality), so one can still take clues from the discernible patterns and regularities of the physical world as the extrinsic appearance of transpersonal mental processes. So the challenge is to find a model of mental computation whose extrinsic appearance would be the laws of physics as we know them. The question is whether cellular automata are suitable for this purpose, as physics doesn't seem to obey e.g. locality or discreteness constraints (not even at the Plank scale).

-- There is a thing called 'digital physics,' which indeed seeks to model all physics with cellular automata. But mental monism should not be portrayed as contingent on the validity of digital physics. Even if physics is non-local (which appears very strongly to be the case), mental monism can still hold. Mentation does not need to be local or discrete for mental monism to hold.

-- I think the author's hypothesis one (discreteness of mind) needs better justification. As it now is, it seems somewhat arbitrary. I, for one, feel that my experience of colors, sounds, fear, love, etc. span a continuous spectrum. On what basis should I believe or hypothesize otherwise? Perhaps the author can argue for discreteness as a useful approximation, instead of arguing that mentation is itself discrete.

-- The author assumes non-spatiality, but still grants the existence of time (without which there would be no causality). Is this internally consistent? Does it require explicit justification? Kant and Schopenhauer posited that both space and time were phenomena constructed within the subject's own mind, not the thing in itself. So I would expect one either goes all the way, like Kant and Schopenhauer, or simply conveniently accepts the objective existence of space and time, as we normally do. After all, Einstein already showed that space and time are different aspects of one and the same thing, so it seems strange to accept one and reject the other.

-- Some of the difficulties (seeming implausibilities) that arise in the paper stem from non-spatiality. It's effectively impossible for human beings to think in non-spatial terms. One option is to then just grant space and time as epistemic limitations of human beings, and work with them in the model proposed. In this case, two minds could very well 'rub against' one another and transmit causal influences from e.g. meta-mental volition and phenomenal perception. By rejecting space, however, the author cannot do this.

-- I do recognize, however, that by assuming non-spatiality the author gets a chance to exercise a number of interesting hypotheses and thoughts that certainly advance the debate. Under spatial assumptions none of these issues would be salient and, therefore, less opportunity for advancement would be present. So I am ultimately fine with it, despite the critical comments above.

More specific comments:

-- Page 2, line 61: "by make" or "by making"?

-- Page 3, lines 89-90: It is not intuitive to me that there is a finite number of moments in any period of time. My intuition, in fact, tends to the opposite conclusion: an infinite number of infinitely short moments. So the author's statement here doesn't seem supported, unless he is appealing to cognitive psychology. In the latter case, a reference would be called for.

-- Page 3, lines 91-92: it seems arbitrary to say that qualities are discrete, as experience seems to indicate precisely the opposite: the continuity of a rainbow, for instance. Perhaps the author can position discreteness as a close-enough simplification that allows us to progress, as opposed to arguing that things are, in fact, discrete.

-- Page 3, lines 132-133: objective idealism (Chalmers 2018) is also an instance of mental monism. Under objective idealism, there is a world out there, which presents itself to us as physicality. It's just that this world is, essentially, mental. In the author's model, this external world is the meta-mind. But the author should explicitly address how his model accounts for objective idealism, instead of limiting his discussion to Berkeley only.

-- Page 9, lines 405-408: the sentence appears to be malformed, incomplete.

Reviewer 3 Report

I recommend rejection of the paper in its current for. Admittedly, that should be taken with a grain of salt, since I am not in information-theory; I’m a philosopher, so I was only able to evaluate the paper insofar as it bears on philosophical questions. I did feel, however, that the paper contains a number of arguments that clearly fall within philosopy rather than information theory, and that my impression of them is that they tend to be ones that would need a lot more work, and which ignore a lot of relevant work in philosophy.

Moreover, I would also say that the problem the paper is trying to address is not entirely clear to me: it would be very useful, at least, to make some reference to people actually raising this problem, so as to demonstrate that and why it is important to solve. A useful paper to look at in order to situate this paper in relation to the contemporary literature would be Chalmers’ “Idealism and the Mind-Body Problem”, http://consc.net/papers/idealism.pdf. Chalmers addresses a number of worries about the capacity of a mentalistic ontology to adequately explain observed phenomena, but they are generally more specific than the very vague objection that the author here considers. It sounds rather as though the objection the author here considers (“the notion that mental monism requires a deus ex machina”) seems more like a lack of imagination on the part of critics than a definite objection.

Relatedly, the official conclusion arrived at - that a world with a mental substrate could support universal Turing computation - seems unobjectionable. Turing computation is substrate-neutral: why wouldn’t we expect that worlds with any sort of substrate can support it? On the other hand, the claims made in the course of getting there seem very bold indeed (‘there is only one subject’!). So that seems rather like burying the lede. I might suggest that a better conclusion to work towards would be either 1) mental monism could generate a pattern of ‘physical’ observations matching those which we in fact have, using explanatory models no less simple than those of physics, or 2) mental monism could generate the subjective experiences that humans in fact have, using explanatory models no less simple than those of neuroscience/scientific psychology.

Arguing for the second of those claims, however, might draw the author into conversation with existing work by panpsychists on the ‘combination problem’, which essentially is exactly the question of how a basic ontology of simple minds could give rise to the sort of minds that humans exhibit. Much of that literature recapitulates some of the discussions that the author has here - for example, the author’s discussion of ‘Hypothesis 6: elementary experientia’ is clearly grappling with what panpsychists call ‘the palette problem’, and suggesting a ‘small palette solution’ (see e.g. Chalmers “The Combination Problem for Panpsychism”, Roelofs “Phenomenal Blending and the Palette Problem”, and Coleman “Panpsychism and Neutral Monism: How to Make Up One’s Mind”).

I see the author has argued in other work that panpsychism and mental monism are not really that different; I think many panpsychists are inclined to agree. But there are some points where I think might put things rather differently from the author. One is that they might understand ‘naturalism’ rather differently. The author takes it to rule out causation which is not ‘local’, and as a result has a lot of trouble making sense of causal interaction once they have themselves pointed out that space is not part of their fundamental ontology. I would suggest that naturalism should not be understood in terms of locality, but in terms of deference to natural science within (some conception of) its proper domain. In particular, I’d think a version of mental monism is ‘naturalist’ if the causal structure is postulates is the same as the causal structure revealed by natural science, e.g. in all causal interactions being explicable ultimately in terms of four basic types of causal interaction between very simple entities. I cannot tell whether the author intends their model to be naturalistic in this sense. Their passing references to god-like ‘metaminds’ is not illuminating on this score, and their frequent references to Berkeley suggests not.

A few remarks about more specific parts of the model presented.

One is that issues of individuation loom very large (to the point of sometimes seeming like the main topic of the paper), but their treatment is based on a starting assumption that seems to me simply false: “Conventionally, it is held that two minds are distinct by virtue of their sitting inside distinct brains, which are in different places… Therefore, [given the unreality of space] subjects simply have no individuation. There is therefore only one subject.”

It’s just not true that the conventional distinction among minds or subjects (or persons, or whatever term we use) is drawn spatially or in reference to brains. It is not clear that there is a single canonical ‘convention’, but in philosophy an extremely widespread view is that minds/subjects are individuated by psychological relations like memory, access, coherence of content, etc. (More info could be found here: https://plato.stanford.edu/entries/identity-personal/#PsyApp). Other options are available, many of them compatible with mental monism. So there is no simple argument here for a single cosmic subject.

(What is particularly odd is that the author seeks to appeal to precisely this sort of psychological structure criterion of individuation later on, in discussing the individuation of ‘minds’ under hypothesis 5. I can’t work out why they don’t take this to also work for the individuation of subjects.)

When discussing hypothesis 1, the author declares that our experiential field (or at least our visual field) must be divided into discrete parts, because we have a maximum acuity. It’s true that we have a maximum acuity - we can’t make arbitrarily fine discriminations. However, that doesn’t by itself show that the field is discrete. In particular, the fact that indiscriminability is generally not transitive seems to problematise such a claim (that is, I might be able to distinguish points A and C, and to distinguish points B and D, but unable to distinguish A from B, B from C, or C from D). Consider this paper for a start: https://www.jstor.org/stable/40271061?seq=1

In discussing hypothesis 3 the author seems to treat volition as equivalent to indeterminism. But indeterminism does not obviously provide room for the will to cause anything (if the will causes, then the result has a cause; it is not undetermined). And there could be volitions that cause things while being themselves fully determined by prior causes.

In discussing hypothesis 4, the author claims that all experiences are located somewhere in experienced space, which is far from uncontroversial; they seem to recognise that it is controversial, and simply one hypothesis out of many, but then I am unclear why they need to make so many strong assumptions in the course of their model. Their proposal for how to construct this space also seems to me to have some holes: for one thing, they take for granted that ‘memory’ allows for the past experiences to be incorporated into present ones, but how does this work? If past experiences can be incorporated into present ones in this way, why not simply allow multiple present experiences to be linked together, without the need for memory? How does this proposal explain the fact that I can, seemingly, open my eyes and immediately see a spatial scene - surely I have not immediately saccaded to every single point (and if I only need a few saccades, that suggests that each one must yield information about multiple points, which seems to presuppose spatial representation itself, which was meant to be constructed…)

In short, each page-long section is addressing a question about which multiple papers have been written, none of them referenced here, and defends an answer to that question for reasons that to me seem unclear or unpersuasive. I do not say this to discourage the author: the paper is a bold effort at exploring some interesting questions. But to be publishable as a philosophy paper it needs to explore those questions in conversation with others who have tried to do the same, much more than it does at present. As an information theory paper, I do not know.

Round 2

Reviewer 1 Report

My previous comments have been adequately addressed.  This is not to say that I agree with all of the points made, but at this point further discussions should be public, not anonymous.

Author Response

I thank the referee for his/her remarks in the first round, and acceptance of the revisions in the second round.

Reviewer 2 Report

Comments on the author's response:

-- Surely our perceptual apparatus has limited resolution, so eventually a digital simulation won't be distinguishable from actual perception, yes. But the intuition behind such thought experiment is that our perceptual apparatus isn't precise enough to pick out the differences. On the other hand, the continuity of the physical world persists even when observations are made through instrumentation, which overcomes perceptual limitations. Would a digital simulation be distinguishable from the physical world if we could also probe it thorough instrumentation? Arguably it very well would, and that's the salient point.

-- The no-communication theorem of quantum information theory, which the author indirectly alludes to in his response, states only that information cannot be transmitted through quantum entanglement. And information non-locality would indeed violate relativity, as the author correctly points out. But the author generalizes this to non-locality in general, which is fallacious and reflects a misunderstanding of quantum mechanics. The only physical interpretations of quantum mechanics still on the run today after the confirmed violations of Bell's and Leggett's inequalities are based on global hidden variables, which are non-local by definition and lead to non-local causal influences. My point thus stands.

-- The author points out that continuity is essentially notional and therefore can be ignored. But the same point, of course, can be made about most other related claims. The discreteness of the world is also essentially notional, for we experience a smooth, continuous world, not a truncated one (whether we can count an infinite number of shades of grey or not). I still think the author should more carefully discuss the suitability of his discrete approach in modeling reality.

-- Digital physics is so directly related to the author's ideas that I think it is inevitable to at least acknowledge it in the manuscript and indicate what is in common, as well as what is different, in the author's approach. It is not obvious that mental monism connects to it (certainly not any more than to any other theory in physics), but it is pretty obvious that the author's model does connect to it.

-- The author disputes my assertion that our perceptions have a feel of continuity, which is rather surprising. The way he disputes it in his response seems silly to me: he asks if I can assert that I distinguish an infinity of shades of grey. Of course I can't assert this, for this requires counting to infinity. My point is one of intuition. Most people will say e.g. that movement is continuous, not a discrete series of instantaneous, truncated steps. It is the author's burden to justify to these people why a discrete model of the world is nonetheless applicable; not my burden to count to infinity.

-- The author insists that the finitude of moments in any period of time can be established merely by appeal to experience and intuition, which seems to fly in the face of experience and intuition. He bases this claims on the same notion that we cannot count to infinity (which is surely true, but that's not the issue in contention). I think this isn't enough. At least since Newton and Leibniz invented the calculus in the 17th century, we've known that any finite interval of space or time can comprise, at least for modeling purposes, and infinitude of infinitely small segments. That's what the calculus notion of limits is all about, and we study that in high school. And since our experience of time is most definitely that of continuous flow, the burden is on the author to justify why his discrete approach is nonetheless applicable.

In general, the author has chosen to dispute nearly all my suggestions for improvement -- except for the trivial language corrections -- as opposed to investigating in a little more depth what is behind them, and adjusting the paper accordingly. I think this is a missed opportunity for improvement, and think the author should thus get another chance to make these improvements.
